# A telomere-targeting drug depletes cancer initiating cells and promotes anti-tumor immunity in small cell lung cancer

Buse Eglenen-Polat[1,2,10], Ryan R. Kowash[1,2,10], Hai-Cheng Huang[1,2], Silvia Siteni[3], Mingrui Zhu[1,2], Kenian Chen [4], Matthew E. Bender [1,2], Ilgen Mender[3], Victor Stastny[5], Benjamin J. Drapkin[2,5], Prithvi Raj[6], John D. Minna [2,5,7,8], Lin Xu [3,9], Jerry W. Shay [2,3] & Esra A. Akbay [1,2] ✉

There are few effective treatments for small cell lung cancer (SCLC) underscoring the need for innovative therapeutic approaches. This study focuses on exploiting telomerase, a critical SCLC dependency as a therapeutic target. A prominent characteristic of SCLC is their reliance on telomerase activity, a key enzyme essential for their continuous proliferation. Here we utilize a nucleoside analog, 6-Thio-2'-deoxyguanosine (6TdG) currently in phase II clinical trials, that is preferentially incorporated by telomerase into telomeres leading to telomere dysfunction. Using preclinical mouse and human derived models we find low intermittent doses of 6TdG inhibit tumor growth and reduce metastatic burden. Anti-tumor efficacy correlates with a reduction in a subpopulation of cancer initiating like cells (CICs) identified by their expression of L1CAM/CD133 and highest telomerase activity. 6TdG treatment also leads to activation of innate and adaptive anti-tumor responses. Mechanistically, 6TdG depletes CICs and induces type-I interferon signaling leading to tumor immune visibility by activating tumor cell STING signaling. We also observe increased sensitivity to irradiation after 6TdG treatment in both syngeneic and humanized SCLC xenograft models both of which are dependent on the presence of host immune cells. This study underscores the immune-enhancing and metastasis-reducing effects of 6TdG, employing a range of complementary in vitro and in vivo SCLC preclinical models providing a potential therapeutic approach to SCLC.

Small cell lung cancer (SCLC) accounts for 13% of lung cancers and is one of the leading causes of cancer-related mortality in the United States with 30,000 deaths annually[1,2]. SCLC is considered a recalcitrant disease by the US Congress and National Cancer Institute with few, if any, current effective therapeutic options. The combination of platinum with etoposide was the primary therapy of SCLC until recently. Most tumors initially dramatically respond to combination therapy with platinum and etoposide but nearly all eventually develop resistance with relapses leading to patient deaths. The combination of immune checkpoint blockade (ICB) with chemotherapy is also FDA approved for use in the treatment of small cell lung cancer (SCLC); however, this combination only provides a few months' benefit when compared to the durable responses frequently observed in non-small cell lung cancer (NSCLC)[2-4]. Low PD-L1 and MHC1 expression, low level of ligands for NK cell activating receptor NKG2D and paucity of immune cells in

the tumor microenvironment (TME) are some of the likely contributors to immunotherapy resistance in SCLC[5–9].

Proteomic profiling studies identified upregulation of DNA repair proteins such as poly-ADP ribose polymerase (PARP1), enhancer of Zeste Homolog 2 (EZH2) and checkpoint kinase 1 and 2 (CHK1/2) in SCLC[10,11]. PARP1 detects single stranded DNA breaks and promotes recruitment of DNA repair proteins to the damaged sites[12,13]. CHK1 and CHK2 sense DNA damage and interrupt cell cycle progression[12,14]. Targeting DNA damage responses with PARP1 and CHK1 inhibitors were effective treatments that also enhanced the effects of immune checkpoint blockade in preclinical studies[15–17]. Induction of DNA damage can activate anti-tumor immunity through Stimulator of interferon genes (STING) signaling[16,18]. Detection of DNA in the cytoplasm by cyclic GMP-AMP synthase (cGAS) leads to activation of STING and Tank Binding Kinase 1 (TBK1), which results in the activation of interferon regulatory factor 3 (IRF3). IRF3 mediates transcription of several immune stimulatory genes including type 1 interferon (IFN)s[19]. While there were early positive signals with PARP inhibitors, alone or in combination with temozolomide in phase 1 trials[20–22], these inhibitors are not yet FDA approved for the treatment of SCLC.

SCLC is a highly metastatic tumor that tends to recur after the initial response to chemotherapy. The potent metastatic capacity and propensity for recurrence observed in SCLC suggest the abundant presence of cancer-initiating cells (CICs) within the tumor tissue[23,24]. CICs in lung cancer are identified by CD133, CD44 and ALDH-1, and importantly telomerase activity[24,25]. Another cancer stem cell or cancer initiating cell marker, L1CAM, is associated with poor prognosis in NSCLC and immunosuppression in pancreatic cancer[26–28]. However, the functional role of L1CAM in SCLC mouse models has not been investigated.

SCLC like most tumors are dependent on the telomerase holoenzyme to bypass replicative senescence resulting from telomere shortening. While 85–90% of all human cancers are telomerase positive, telomerase expression is not detected in most human adult tissues except for proliferating stem-like cells[29]. SCLCs are nearly all telomerase positive[30], suggesting that targeting telomerase maybe a reasonable strategy in the treatment of SCLC. The first in class direct telomerase inhibitor GRN163L (Imetelstat) showed in vitro efficacy against NSCLC[31] but did not improve progression free survival (PFS) in clinical trials including trials with advanced NSCLC patients[32]. The clinical results in retrospect are not surprising since cancer cells treated with Imetelstat have to undergo many rounds of cell division for their telomeres to reach critical point leading to cell death[33]. In contrast, 6-Thio-2′-deoxyguanosine (6TdG), a modified purine analog with high-affinity for telomerase has anti-tumor activity independent of the telomere length leading to a significantly shorter onset of action compared to Imetelstat[34]. Upon in vivo administration, 6TdG is converted to 6-thio-2′-deoxyguanosine-5′-triphosphate which is immediately incorporated into telomeres, causing rapid telomere dysfunction and uncapping, resulting in activation of DNA damage responses and apoptosis in tumor cells[34]. 6TdG was shown to be effective in NSCLC[34,35], NRAS-mutant melanoma[36] and pediatric brain tumor models[37]. In an immunogenic colon cancer model, this inhibitor proved to be acting primarily through dendritic cells to activate anti-tumor immunity[38].

In this study, we present evidence for the anti-tumor efficacy of 6TdG in both chemo-naïve and resistant patient-derived SCLC lines, syngeneic immunocompetent mouse models, and humanized SCLC xenograft models. Our findings indicate that 6TdG effectively hampers primary SCLC tumor growth and mitigates metastasis, achieving this by inducing telomere dysfunction and thereby selectively depleting cancer-initiating cells (CICs) which are characterized by high telomerase activity. Furthermore, our in vivo experiments indicate that the therapeutic impact of 6TdG involves NK and CD8 T cells, as evidenced by a diminished therapeutic response in mice lacking functional NK or CD8 T cells. Notably, when combined with radiotherapy, 6TdG exhibits durable responses, suggesting a promising therapeutic strategy for SCLC.

## Results

### Human and mouse SCLC lines are sensitive to 6TdG

Currently, cisplatin in combination with etoposide is the standard chemotherapeutic regimen used in SCLC treatment[2–4]. We evaluated the efficacy of 6TdG in comparison to cisplatin. SCLC cells tested—human H1048, H69, and H510 (Fig. 1a) and mouse SCLC cell line-984 were slightly but significantly more sensitive to 6TdG than cisplatin. 984 cells were developed from a conditional loss of Rb and p53 transgenic mouse model of SCLC. Sensitivity to 6TdG treatment in vitro was analyzed in additional mouse SCLC cell lines 984 A (adherent, variant SCLC), and RPP (*Rb, p130*, and *p53* triple knockout) (Fig. 1b). 6TdG treatment inhibited growth of mouse and human small cell lung cancer cells, with IC50s ranging between ~0.2uM and ~1.0uM for human cells and ~0.2 and ~0.9 for murine SCLC cells (Fig. 1a, b). To determine whether 6TdG is effective in the chemoresistant setting, we utilized an SCLC patient derived xenograft model (PDX). This model was derived from an SCLC patient who received etoposide, temozolomide and topotecan[21]. We previously showed that this PDX model maintains chemo- resistance in vitro and has MYC amplification, which has been reported to contribute to chemoresistance in SCLC[39,40]. This PDX model was sensitive to 6TdG treatment with an IC50 of ~0.6 and ~0.8 uM in 2 independent passages (Fig. 1c). Although loss of neuroendocrine differentiation is associated with chemo-resistance in SCLC, non-neuroendocrine SCLC lines were also sensitive to 6TdG treatment, as evidenced by the decreasing number of colonies with increasing doses of 6TdG in colony formation assays with mouse (984 A) and human adherent SCLC cells (H1048, H841) which express a non-neuroendocrine phenotype (Fig. S1a). We compared the efficacy of 6TdG to the direct telomerase inhibitor, Imetelstat. We observed that Imetelstat has no cytotoxic effects in five-day in-vitro in IC50 assays in mouse SCLC cell lines 984 and RPP and human SCLC cell lines H69 and H510 (Fig. S1b). We tested the efficacy of imetelstat in vivo in a 984 flank syngeneic tumor model and observed no significant decrease in tumor growth. (Fig. S1c).

In addition to reduced viability with 6TdG, we also observed increased cellular senescence when cells were treated with 6TdG in vitro. Adherent SCLC cell lines 984 A, H841, and H510 were treated with either 0.5 uM or 1 uM 6TdG for 48 h, and then incubated for 5 additional days without the drug. Using the beta galactosidase senescence assay, we observed a significant increase in senescent cells across both doses of 6TdG in all three cell lines (Fig. 1d). Positive cells are defined by the presence of blue color signifying the detection of beta gal, which is a commonly used senescence marker (Fig. S1d). We also observed an increase in gamma-H2AX (gH2AX) positive cells among live 984 cells with 6TdG treatment, indicative of increased DNA double stranded breaks with the treatment (Fig. 1e). To determine the localization of the double stranded breaks, we performed a telomere dysfunction induced foci (TIF) assay. The TIF assay utilizes co-localization of a DNA damage marker such as g-H2AX with one of the telomere capping proteins such as TRF2[41]. Human SCLC cell line H841 was treated with 6TdG for 48 h before staining. There was increase in both general double stranded breaks and telomere specific breaks indicated by g-H2AX accumulation colocalized with telomere probe Tel-C within the nucleus (Fig. S2a). The numbers of TIFs and g-H2AX- foci per cell were 31 or 19 times higher for cells treated with 6TdG as compared to controls respectively (Fig. S2b). While there is damage outside of telomeres too, since telomeres are only 1/6000th of the genomic DNA, these data indicate preferential enrichment of damage at the telomeres. 6TdG treatment also resulted in formation of various chromosome defects such as single fusion chromosomes at telomere sites, dicentric chromosomes fused at their telomeres, and free telomere fragments, as a result of telomere deprotection (Fig. 1f).

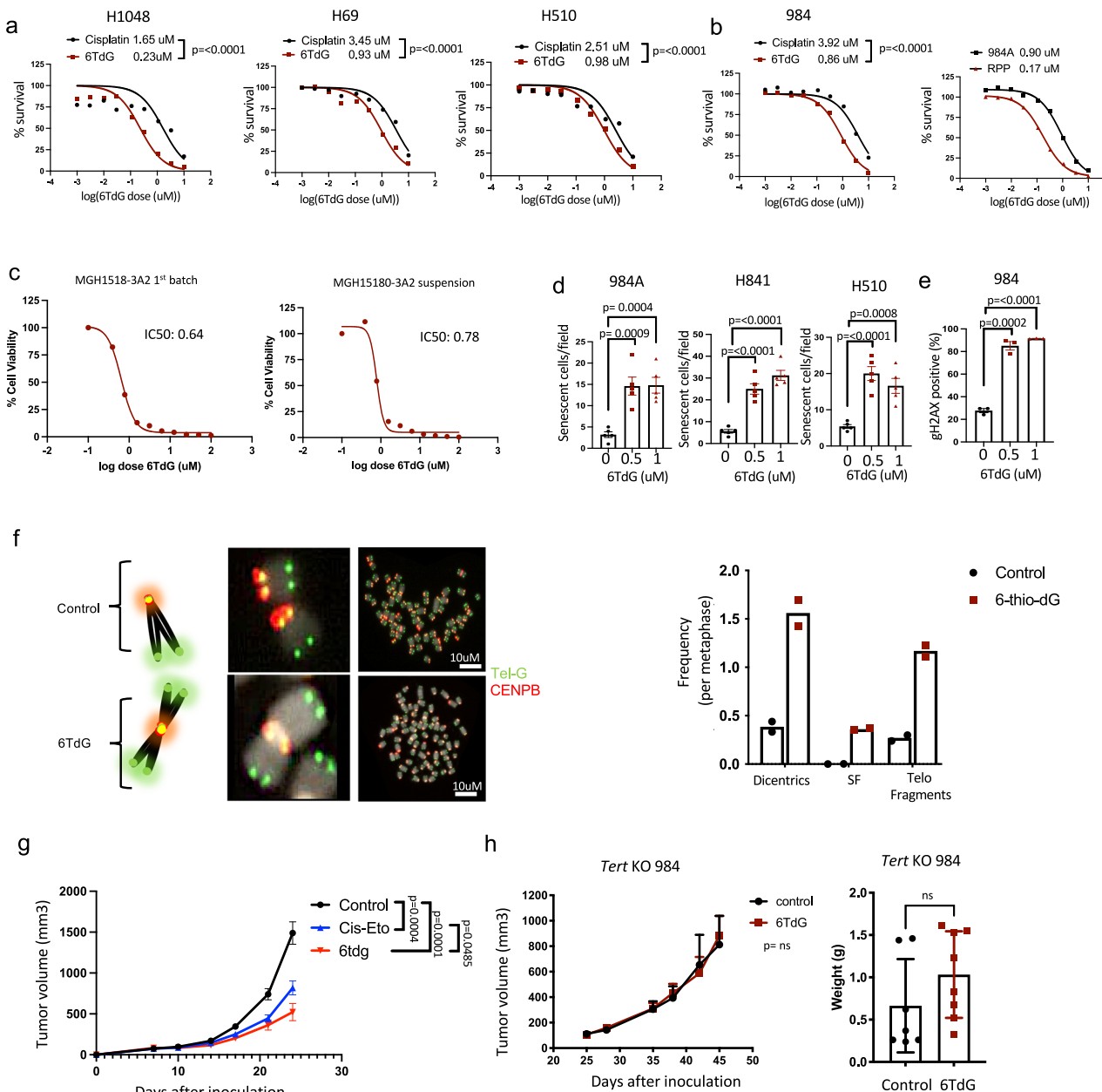

**Fig. 1 | Human and mouse SCLC lines are sensitive to 6TdG. a** Cell viability curves and IC50 values of human SCLC cell lines H1048, H69, and H510 treated with cisplatin or 6TdG. Best fit curves were compared for statistical analysis. *n* = 3 per group. **b** Cell viability curves and IC50 of cisplatin or 6TdG on murine SCLC 984. Best fit curves were compared for statistical analysis. *n* = 3 per group. Cell viability curves and IC50 values of mouse SCLC cell lines 984 A, and RPP treated with 6TdG(right). *n* = 3 per group. **c** Cell viability curves and IC50 values of human SCLC PDX MGH1518.3A2 1st passage or suspension treated with 6TdG. *n* = 3 and 6 replicates per group respectively. **d** Counts of senescent cells determined by senescence associated beta-gal assay per microscope field of view of mouse 984 A and human H841 and H510 SCLC cell lines treated with vehicle, 0.5 uM, and 1.0 uM 6TdG. *n* = 5 samples per group. Error bars represent mean + standard error of the mean (SEM). Groups were compared by unpaired *T* tests. **e** Flow cytometry analysis of gH2AX + cells (% of live cells) in 984 cells treated with vehicle, 0.5 uM or 1.0 uM

6TdG for 48 h. *n* = 3 for each group. Error bars represent mean + SEM. Statistical analysis was completed by unpaired *T* tests. **f** Chromosome spreads showing chromosomes of control or 3 uM 6TdG treated 984 cells (48 h). Representative images of a dicentric chromosome enlarged on the left. Quantification of chromosome defects is shown on the right. **g** Growth curve for subcutaneous 984 tumors implanted in WT B6129F1 mice and treated with vehicle (*n* = 12), cisplatin and etoposide (*n* = 12) or 6TdG (*n* = 7). An unpaired *T* test was used for statistical analysis of the last measurements. Error bars represent mean + SEM. **h** Growth curves of *Tert*-KO 984 cells implanted into WT B6129 mice subcutaneously and treated vehicle or with 6TdG (*n* = 8 for both, left). Final tumor weights of *Tert*-KO subcutaneous tumors (*n* = 7 for vehicle and 8 for 6TdG, right). Tumor volumes and weights were compared with unpaired *T* tests. Error bars represent mean + SEM for tumor growth curves and mean + SD for tumor weights. Source data are provided as a Source Data file.

To recapitulate the efficacy of 6TdG on an in vivo SCLC model and to compare this to chemotherapy, we treated 984 flank tumors with 6TdG or the standard clinically used combination of cisplatin and etoposide. Both treatments were effective but 6TdG treated tumors were significantly smaller at the end point compared to chemotherapy

combination (Fig. 1g). To establish that 6TdG was acting specifically through telomerase, we investigated the effect of 6TdG on *Tert* knockout 984 cells. Because mice have very long telomeres it is possible to get mouse *Tert* knockout tumors to grow, albeit slower than wildtype tumors. Importantly, the growth of 6TdG treated *Tert*$^{-/-}$ 984

flank tumors were comparable to the vehicle treated controls (Fig. 1h) indicating that therapeutic action of 6TdG in vivo is dependent on the continued expression of telomerase in tumor cells.

## 6TdG decreases cancer initiating cells and diminishes tumor initiation potential in vitro and in vivo

We hypothesized that cancer initiating cells (CICs) may be preferentially sensitive to 6TdG treatment due to their putatively higher levels of telomerase activity compared to all cells in a tumor. We searched for a potential marker to track these cells for these studies. L1 cell adhesion molecule (L1CAM) was shown to be associated with metastasis[42] and drug resistance[43] in cancer patients and L1CAM positivity was associated with a higher recurrence rate[44]. CD133 is another marker of stemness in SCLC and is found to be associated with chemotherapy and radiotherapy resistance[45–47]. Analysis of previously published SCLC datasets from primary tumors[48] and cell lines[49] indicate that the majority of SCLCs express either CD133 (PROM1), L1CAM or both at the mRNA level (Fig. 2a). Thus, we investigated the differences in L1CAM/CD133 positivity in SCLC cell lines and mouse tumors in response to 6TdG. L1CAM+ cell populations had higher Ki67 positivity as detected by flow cytometry ($p = 0.0178$) (Fig. 2b). Murine and human SCLC cell lines have variable baseline expression of L1CAM, (76%, 14%, and 49% positivity; for 984, H2081 and H841 cells, respectively) and flow cytometry analysis revealed significant dose-dependent reduction in L1CAM positivity in 6TdG treated cells (Fig. 2c). 6TdG treatment also significantly reduced CD133 positivity in H69 cells (Fig. 2d). When treated with 6TdG, L1CAM positive cells had a higher level of annexin V as compared to L1CAM negative cells (Fig. S3a), indicating that L1CAM+ cells are more sensitive to 6TdG treatment.

Relative telomerase activities of different SCLCs were determined using a real-time quantitative PCR based assay relying on telomerase in the tumor cell lysate and telomere primer set in the kit amplifying newly synthesized telomere sequences. As predicted, L1CAM+ or CD133 high subsets of SCLC cells had higher level of relative telomerase activity as compared to L1CAM- or CD133 low (984, H2081, H69) cells (Fig. 2e). When separated via FACS (Fig. S3b, d, e) and cultured, L1CAM + 984 or L1CAM and CD133 high H69 cells grew larger 3D clusters than their L1CAM-, CD133 low counterparts (Fig. 2f). In leukemia, CICs were shown to be less visible to the immune cells by reduction in NK cell activating ligands[50]. Thus, we looked for a similar phenotype in SCLC and found L1CAM positive 984 cells expressed lower levels of MHC-1 as compared to the L1CAM negative tumor cell population (Fig. 2g). L1CAM function as a marker of CICs was also confirmed in vivo. When 984 cells were sorted according to high and low L1CAM expression via FACS (Fig. S3c) and injected into wildtype mice, L1CAM high cells developed significantly higher number of metastatic lesions as compared to L1CAM low 984 cells (Fig. 2h). This was reflected as a higher metastatic tumor burden in L1CAM high injected cells as compared to L1CAM low cells as indicated by liver/body weight ratios (Fig. 2h).

## Low dose 6TdG is effective in treating metastatic mouse SCLC tumors

To further investigate the in vivo efficacy of 6TdG, we used the syngeneic 984 SCLC model that reproduces the aggressive and metastatic nature of SCLCs in patients. When these cells are intravenously (IV) injected, they primarily colonize the livers and lungs, with mice dying from large tumors in the liver[8]. We confirmed presence of liver tumors 10 days after IV injection of tumor cells by H&E staining of tumor tissue (Fig. S4a). Mice were then treated with 4 intermittent doses of 3 mg/kg 6TdG or vehicle (5% DMSO solution) (Fig. 3a). 6TdG significantly decreased overall tumor burden in the livers as shown by ratios of liver weight to body weight (Fig. 3b). The number of SCLC lesions and tumor areas in livers were also significantly smaller in treated mice

(Fig. 3b). We performed immunohistochemistry against neural transcription factor ASCL1 expressed by 984 cells to be able to quantify microscopic lung lesions. The number of ASCL1+ lesions in the lungs were significantly less in the treated mice (Fig. 3c). To demonstrate the in vivo pharmacodynamic effects of 6TdG on metastatic SCLC, wild type mice were injected IV with 984 cells and treated per the short-term schedule (Fig. 3d). Proliferation index significantly decreased in tumor areas in treated mice as indicated by Ki67 immunohistochemistry (IHC). (Fig. 3e). Cleaved caspase 3 and gH2AX, markers of apoptosis and double stranded DNA breaks, respectively, were significantly increased in the tumor areas of treated tissues, as determined by IHC (Fig. 3f, g). Consistent with the in vitro data, 6TdG led to increased cell death and increased DNA damage in mouse tumors.

We also evaluated the efficacy of 6TdG in the murine RPP SCLC model derived from *Rb; Trp53; p130* conditional mutant mice[8]. When injected subcutaneously, this model metastasizes to the liver of the mice. Four intermittent doses of 6TdG treatment significantly inhibited growth of RPP primary flank tumors in nude mice ($p = 0.0014$, Fig. 3h, i). Notably, at the time of euthanasia when controls reached endpoint, metastatic burden was significantly lower in the 6TdG treated mice as compared to vehicle treated mice ($p = 0.0412$, Fig. 3j). We point out that we do not know the exact step of metastasis inhibited by 6TdG.

## CD8 and NK cells contribute to the efficacy of 6TdG in vivo

To investigate the cellular dependencies of 6TdG response, we performed therapeutic experiments by depleting CD8 or NK cells in our syngeneic murine 984 metastatic model. Depletion efficiency of T and NK cells by antibodies was established by flow cytometry (Fig. S4b). Depletion of cytotoxic T cells or NK cells with an antibody against CD8T cells (CD8a) or NK cells (NK1.1) significantly reduced the efficacy of intermittent 6TdG treatments in metastatic models, as compared to 6TdG treatment alone (Fig. 4a–c). This data demonstrates that in this immunocompetent mouse SCLC model, both CD8 T cells and NK cells contribute to the in vivo anti-tumor response to intermittent 6TdG therapy.

To identify potential immune modulating effects of 6TdG on SCLC cells several obvious markers were evaluated. 6TdG treatment induced SCLC MHC1 expression as determined by flow cytometry (Fig. 4d). Additionally, expression of NK cell activating ligands NKG2DLs, specifically RAE1d isoform, and DNAM1 ligands are significantly increased in murine 984 SCLC line treated with 6TdG (Fig. 4e). Similar trends were observed in human SCLC lines. 6TdG treatment increased MHC-1, DNAM-1Ls, NKG2DLs (Fig. 4f). Interestingly, 6TdG also induced NK cell inhibitory ligands HLA-E and HLA-G expression on H69 and H510 lines (Fig. S4c).

To evaluate the impact of these changes on the tumor cell surface on T or NK cell activation, we utilized co-culture assays with healthy donor PBMCs and NK cells with 6TdG or control pretreated H69 or H510 cells. T cells exhibited higher cytotoxicity against 6TdG pretreated H510 as compared to vehicle treated cells, as indicated by increased production of Granzyme B and IL-2 on CD8 T cells using flow cytometry (Fig. 4g). Secreted Granzyme B and IFN-gamma were significantly higher in PBMCs co-cultured with pretreated H510 cells as using ELISA (Fig. 4g). A similar trend was observed with Granzyme b expression and secretion with PBMCs co-cultured with H69 cells. To enrich lymphocytes from the PBMCs we isolated CD3 + T cells for co-cultures (Fig. S4d). CD3+ PBMCs co-cultured with 6TdG pretreated H510 cells also demonstrated significantly higher Granzyme B, IFN-gamma, and IL-2 surface expression by flow cytometry (Fig. S4e). Co-culture of 6TdG pretreated H510 cells with CD4+ PBMCs also led to increased IL-2 and IFN-gamma production in CD4 + T cells (Fig. S3f). Similarly, when pretreated H510 cells were co-cultured with the human NK cell line, NK92, we observed increased tumor cell lysis and increased NK cell

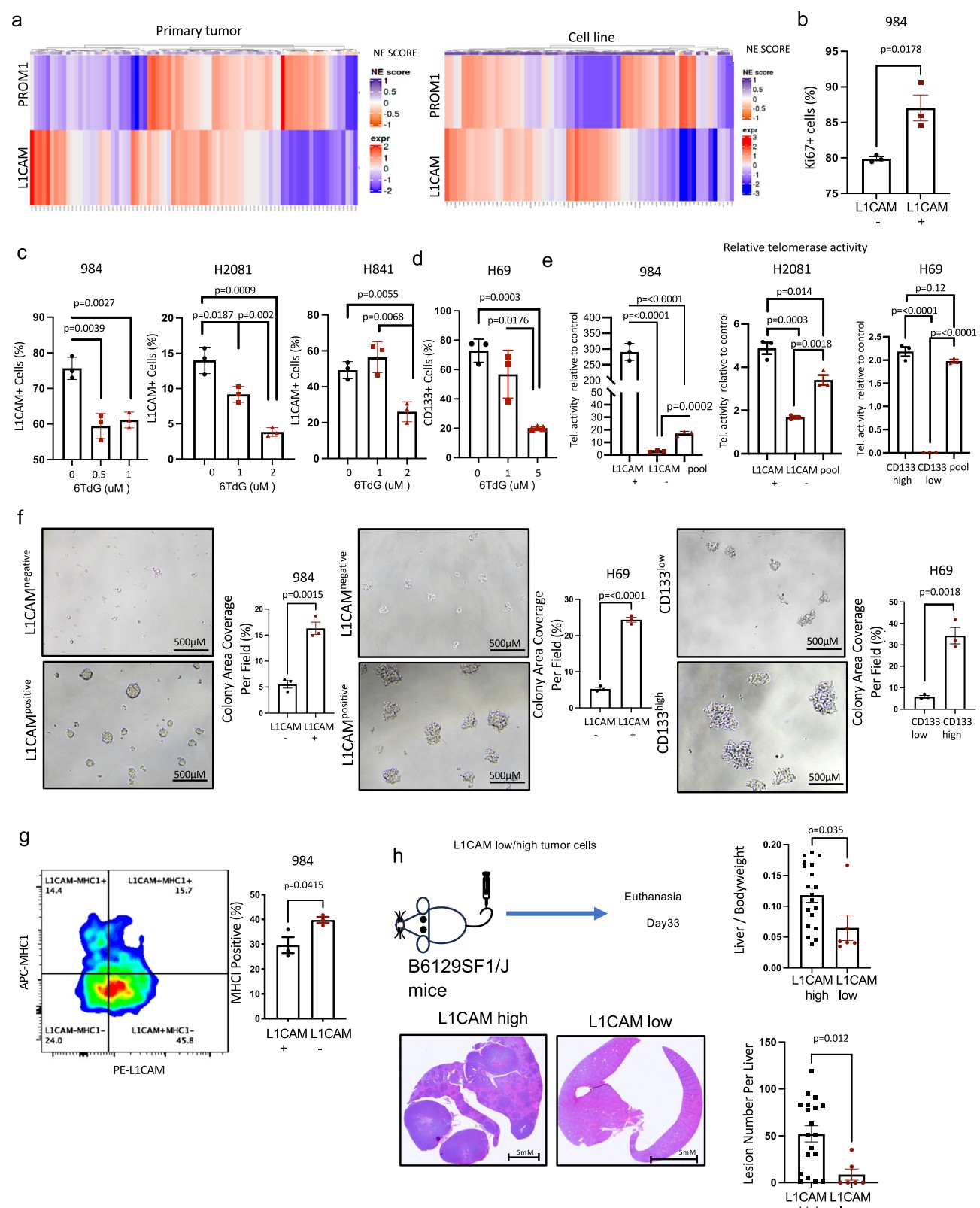

activation in 6TdG pre-treated cells as compared to control cells ($p < 0.01$) (Fig. S4g). Impact of 6TdG treatment of cancer cells on mouse lymphocytes was similar. Co-culture of mouse splenocytes with 6TdG-pretreated 984 cells resulted in increased Granzyme B expression among CD3 + T cells and increased IFN-gamma production from CD8 + T cells and NK cells (Fig. S4h).

**6TdG activates type-I interferon pathway through cGAS-STING**
We hypothesized that 6TdG treatment and resulting increase in aberrant chromosomes may result in increased cytoplasmic DNA and, in turn, activate cGAS-STING signaling. 6TdG treatment for 24 h induced micronuclei formation in 984 cells (Fig. 5a). Some of the micronuclei are coated with nuclear envelopes, and some of them

**Fig. 2 | 6TdG decreases tumor initiating cells and diminishes tumor initiation potential in vitro and in vivo. a** Heatmaps displaying expression of *L1CAM* or *PROM1* (CD133) in human SCLC primary tumors and cell lines, original data from ref. 48 and ref. 49. Data is z-transformed for each gene with the scale shown on the right. Neuroendocrine (NE) score is also indicated. **b** % of Ki67 + 984 cells gated as *L1CAM* negative and L1CAM positive cells with flow cytometry (avg 79.9% vs 87.0%). $n = 3$ per group. Error bars represent mean + SEM. Unpaired *T* tests were used for analysis. **c** % L1CAM + 984, H2081, H841 cells treated with vehicle or 6TdG as determined by flow cytometry. $n = 3$ for each group. Error bars represent mean + SEM. Unpaired *T* tests were used for analysis. **d** % CD133 positive cells in vehicle or 6TdG treated H69 cells at the indicated doses for 5 days analyzed by flow. Error bars represent mean + SEM. Unpaired *t* test was used for analysis. $n = 3$ replicates per condition. **e** Relative telomerase activity (ΔCq) in sorted 984, H2081 and H69 cells compared to unsorted pool. (ΔCq) as determined by qPCR assay based on telomerase in cell lysate and telomere primer set. $n = 3$ for each group. Error bars represent mean + SEM. Groups were compared by multiple unpaired *T* tests. **f** Representative images of 3D clusters of L1CAM negative/positive 984 and H69 cells and CD133 low/high H69 cells plated after sorting. % colony area per field was quantified from the microscope images. $n = 3$ for each group. Error bars represent mean + SEM. Unpaired *T* tests were used for analysis. **g** Representative flow cytometry plot of MHC-1 expression in unsorted L1CAM positive/negative 984 cells (left) and quantification of flow (right). $n = 3$ for each group. Error bars represent mean + SEM. Unpaired *t* test was used for analysis. **h** Schema for in vivo experiment with sorted L1CAM high/low 984 cells in WT mice (top left). Representative liver H&E images of mice (bottom left), liver weight/body weight ratios of mice injected with L1CAM high/low 984 cells ($n = 19$ vs 6) (bottom right). Error bars represent mean + SEM. Unpaired *t* tests were used for statistical analysis. Source data are provided as a Source Data file.

were uncoated and exposed in 6TdG treated cells, while no micronuclei were observed among control cells (Fig. 5a).

To establish activation of STING pathway with the increase in micronuclei numbers with 6TdG treatment, human SCLC cell line H841 was treated with 6TdG for 8, 24 or 48 h. Levels of pSTING and pTBK1 were higher within 24 h and 48 h of 6TdG treatment (Fig. S4i). This data connects 6TdG treatment to the STING pathway. Treatment of human SCLC lines H69 or H2081 with 6TdG induced expression of type-I interferons (IFNa and IFNb) downstream of cGAS/STING pathway. Consistent with IFNs, expression of interferon stimulated genes (ISG)s (TAP1, LMP2, LMP7, CXCL10, and B2M) were also significantly elevated in both the human SCLC lines. Additionally, CCL5 was elevated in H69 while not detectable in H2081 (Fig. 5b). We found this also occurs in mouse SCLC tumors treated with 6TdG, with liver tumors from 6-TdG treated mice showing significantly higher levels of IFNa and IFNb expression using qPCR analyses (Fig. 5c).

To further determine the contribution of cGAS and STING to immune mediated effects of 6TdG we performed genetic inactivation experiments. Increase in the expression of IFNa, IFNb, and ISGs was lost in *Cgas*-KO 984 cells as compared to *Cgas* wt-984 when treated with 6TdG, indicating the role of cGAS in inducing expression of these genes (Fig. 5d). While 6TdG treatment increased surface MHC1 and NKG2DL expression in wild-type H69 cells, this effect was not observed in *STING*-KO H69 cells treated with 6TdG indicating that induction of immune visibility by 6TdG is dependent on tumor STING signaling (Fig. 5e). To further determine the role of tumor STING in the therapeutic efficacy of 6TdG in mice, vector control *or Sting* KO 984 cells were implanted into WT mice by IV and mice were treated with intermittent 6TdG (Fig. 5f). There was no significant difference in the tumor burden between vehicle or 6TdG treated *Sting* KO 984 tumors (Fig. 5g), suggesting that in this SCLC model system the therapeutic effects of 6TdG are dependent on tumor cell STING expression.

## 6TdG treatment modulates the tumor microenvironment

As 6TdG effects were immune cell mediated in mice, we performed an unbiased analysis of the tumor microenvironment through single-cell RNA sequencing. We treated wild type mice injected with 984 cells by IV and treated according to the previously detailed short-term treatment protocol (Fig. 6a) and performed single cell RNA sequencing. We complemented single cell sequencing data with immunophenotyping by flow cytometry and IHC. 6TdG treatment induced recruitment of lymphocytes into the tumor microenvironment (T cells, NKT cells, NK cells and B cells), while reducing subsets of myeloid cells (Fig. 6b and Fig. S5). Overall, 6TdG created an anti-tumor immune environment, associated with increased lymphocytes. Of note, we observed a significant increase in the *TCF7+* cell population among T, NK, and NKT cells in the treated group compared to control (Fig. 6c). This was also confirmed at the protein level by IHC, where we observed increased TCF1+ (encoded by *TCF7* gene) cells in the treated liver tissue (Fig. 6d). TCF1 positive cells are recognized as critical mediators of

response to immunotherapies such as immune checkpoint blockade antibodies[51].

In confirmatory flow experiments, we observed a significant increase in CD45+ cells and a significant decrease in tumor cells (28% vs 5%, $p = 0.0045$) relative to total live cells in the metastatic liver tumor (Fig. 6e). With 6TdG treatment, the percentage of total T cells (CD3+) (mean %: 24% vs 35%) were increased among CD45+ cells, while CD11b + Ly6C+ myeloid derived suppressor cells (MDSC) (mean: 27% vs 7%) were reduced. NK cells (NK1.1 + ) and B cells (CD19 + ) among CD45+ cells were not significantly different between control and treatment groups in this model when quantified by flow (Fig. 6e, Fig. S6a).

Among the T cells, CD4 + (42% vs 50%, $p = 0.0237$) and CD8 + (25% vs 30%, $p = 0.09$) T cells showed a trend towards increase, while double negative (CD4-CD8-) T cells (35% vs 22%, $p = 0.0082$) were significantly reduced in the treatment group (Fig. 6f). 6TdG treatment resulted in significantly reduced expression of checkpoint receptors, PD-1 and Tim3, on CD8 + T cells (PD-1: 8.2% vs 3.5%, p = 0.0097; Tim3: 6.7% vs 3.2%, $p = 0.0097$) (Fig. 6f). FOXP3 positivity among CD3 + T cells (8.0% vs 3.6%) or CD4 + T cells (15.6% vs 7.4%) significantly decreased in 6TdG treated mice compared to control (Fig. 6g). CD4 + FOXP3+ regulatory T cell ratios were also decreased among total T cells (5.6% vs 2.5%) (Fig. 6h) and ratio of CD8+ cells to Tregs which favors T cell activity increased with treatment (7.5 vs 19) (Fig. 6h). In a separate experiment we confirmed activation of T cells after brief treatment with 6TdG. CD8 + T cells from the tumors significantly increased Granzyme b, Ki67, and IFN-gamma expression in the treated mice (Fig. S6b) and CD107a expression was also higher in CD8 + T cells. NK cells were also more activated with treatment. Finally, L1CAM positivity was significantly reduced in the tumor tissues after 6TdG treatment consistent with in vitro experiments (Fig. S6b).

To determine the effects of 6TdG on healthy hematopoietic cells in non-tumor tissues after long-term treatment, we analyzed circulating blood cell counts and bone marrow cells. There were no significant differences in white blood cell (WBC) counts, lymphocyte counts, or percentages of lymphocytes in the blood of mice euthanized 20 days after last treatment with vehicle or 6TdG (Fig. S6c). Colony formation ability of bone marrow cells from long-term 6TdG treated mice was similar to that of vehicle treated mice Fig. S6d). This data supports the idea that 6TdG does not induce bone marrow suppression in this setting.

## 6TdG potentiates irradiation in SCLC

Considering that 6TdG will enter clinical trials in SCLC in the setting of existing therapies, experiments examining the effects of 6TdG in standard of care platin-etoposide resistant cells were initiated. After confirming that 6TdG is effective in chemo-resistant cells, we set out to determine whether the therapeutic effect can be potentiated by radiotherapy (IR) widely used in SCLC treatment that also has potential immunogenic effects[52]. We tested the sensitivity of murine 984 SCLC

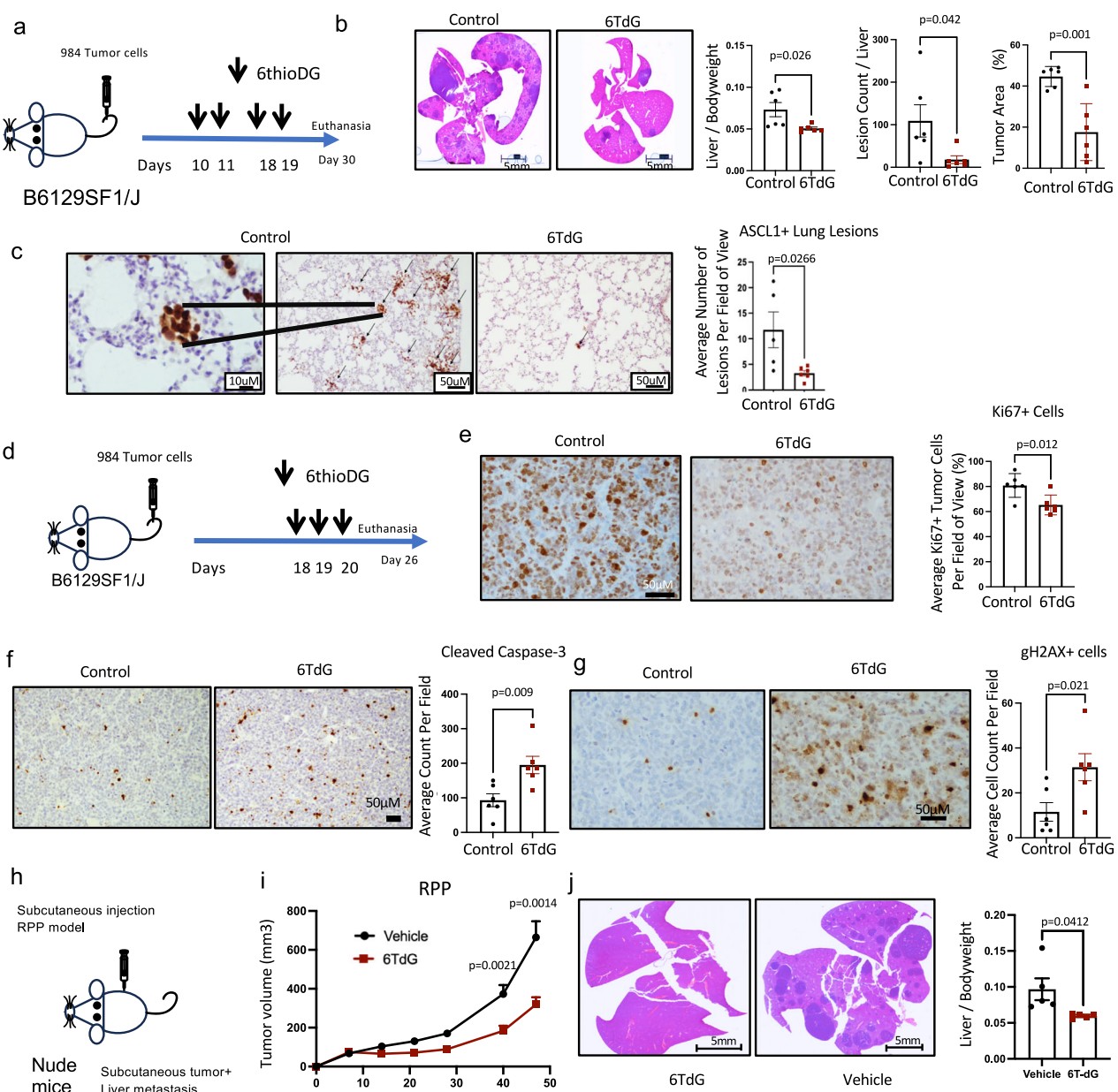

**Fig. 3 | 6TdG induces double stranded breaks and apoptosis in mouse tumors.**
**a** Schema for long-term in vivo study 984 cells injected IV into WT B6129SF1/J mice.
**b** Representative H&E-stained liver sections of vehicle or 6TdG treated mice at the endpoint (left). Liver weight/body weight ratio ($n = 6$ for control and $n = 6$ for 6TdG); lesion counts ($n = 6$ for control and $n = 6$ for 6TdG); and quantification of tumor area ($n = 6$ for control and $n = 6$ for 6TdG) from the H&E sections. Error bars represent mean + SD. Groups were compared by unpaired $T$ tests. **c** Representative images of ASCL1 IHC on the lung tissue from the mice in **a** ($n = 5$) or 6TdG ($n = 6$) (left) and quantification of average ASCL1 positive lesions per field of view (mean lesion counts: 11.7 vs 3.3, right). Error bars represent mean + SEM and statistical analysis completed by unpaired $T$ test. **d** Schema for short term treatment of WT mice injected with IV 984. 6 days after completion of treatment animals were sacrificed for downstream analysis. **e**–**g** downstream analysis of tissues from treated mice. **e** Representative Ki67 staining of liver tumors and quantification of staining (average Ki67 positive tumor nuclei (%). Error bars represent mean + SD.

Control ($n = 6$) and 6TdG treated ($n = 6$). Statistical analysis completed by unpaired $T$ test. **f** Representative images and quantification of cleaved caspase 3 positive tumor cells in vehicle ($n = 6$) or 6TdG ($n = 6$) treated mice. Error bars represent mean + SEM and statistical analysis completed by unpaired $T$ test. **g** Representative images and quantification of average gH2AX positive cells perf field in control ($n = 6$) or treated ($n = 6$) mice (right). Error bars represent mean + SEM. Statistical analysis completed by unpaired $T$ test. **h** Schema for RPP mouse SCLC subcutaneous injection into nude mice treated with vehicle or 6TdG and monitoring of liver metastasis. **i** Growth curves of primary flank tumors in vehicle ($n = 5$) or 6TdG treated ($n = 5$) mice, tumors on both sides of the mice. Error bars represent mean + SEM **j** Representative H&E images of RPP tumors cells that metastasized from the flank to the liver in nude mice. Liver to bodyweight ratio is quantified in vehicle ($n = 5$) or 6TdG ($n = 5$) treated groups. Error bars represent mean + SEM and values compared by unpaired $T$ test. Source data are provided as a Source Data file.

cells to IR after 24 h of 6TdG treatment in vitro. The combination effect was modest in vitro (Fig. 7a). By contrast, the in vivo effect was more pronounced with intermittent and low dose 6TdG significantly increasing the efficacy of one single dose of IR in the syngeneic model

(Fig. 7b). Combination treatment with 6TdG and 5 Gy IR caused significantly reduced tumor growth compared to control group ($p < 0.0001$) and the combination was significantly more potent than either 6TdG or IR treatment alone ($p = 0.0018$ and $p = 0.015$ for

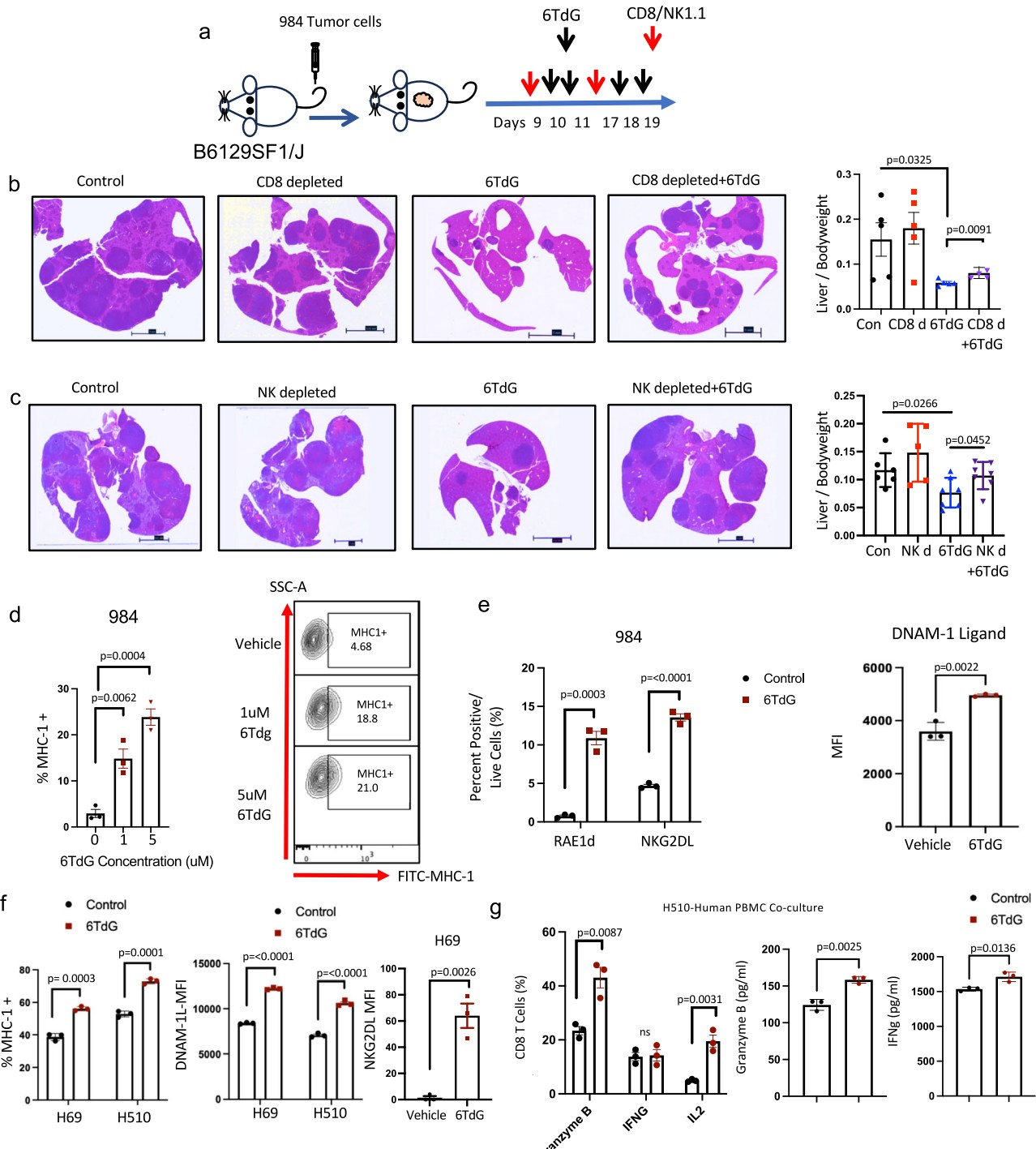

**Fig. 4 | CD8 T cells and NK cells contribute to the therapeutic efficacy of 6TdG.**
**a** Scheme for in vivo study of 984 cells injected by IV into mice and treated with CD8 or NK cell depletion antibodies or 6TdG. Animals were followed long term and sacked when control mice reached endpoint due to tumor burden.
**b** Representative H&E images of tumor-bearing liver of mice treated with vehicle ((*n* = 5), CD8 antibody (*n* = 5), 6TdG (*n* = 5) or 6TdG and CD8 antibody (*n* = 5) (left) and liver/ bodyweight ratios (right). Scale bars show 5 mm for **b** and **c**. Error bars represent mean + SEM. Multiple unpaired *T* tests were used for statistical analysis.
**c** Representative H&E images of liver tissue from vehicle (control) (*n* = 6), NK1.1 (*n* = 5), 6TdG (*n* = 7) or 6TdG and NK1.1 antibody (*n* = 7) (left) and liver/ bodyweight ratio on the right. Error bars represent mean + SD. Statistical analysis was completed by multiple unpaired *T* tests. **d** Quantification of MHC-1 staining (% of live) 984 cells treated with vehicle, or 1 uM and 5 uM 6TdG (left) and representative flow plots for the MHC-1 expression (right). Error bars represent mean + SEM and *n* = 3

per group. Groups were compared by unpaired *T* test. **e** RAE1d or NKG2DL expression (stained with NKG2D-Fc, left) and DNAM-1 (stained with DNAM1-Fc) expression in vehicle (*n* = 3) or 6TdG (*n* = 3) treated 984 cells (right) after five day treatment. Error bars represent mean + SEM. Data was compared by unpaired *T* test. **f** MHC1 expression (left) and DNAM-1L expression (middle), and NKG2DL expression in H69 cells treated with vehicle (*n* = 3) or 6TdG (*n* = 3, right). Error bars represent mean + SEM. Data was compared by multiple unpaired *T* tests. **g** Healthy donor PBMCs were co-cultured with vehicle or 6TdG pre-treated H510 cells. The ratio of CD8 T cells expressing Granzyme b, IFNg, and IL2 analyzed by flow (left) and Granzyme b and IFNg from the co-cultures of vehicle or 6TdG pre-treated tumor cell groups quantified by ELISA (right, *n* = 3 per group). Error bars represent mean + SEM. Statistical analysis was completed by multiple unpaired *T* tests. Source data are provided as a Source Data file.

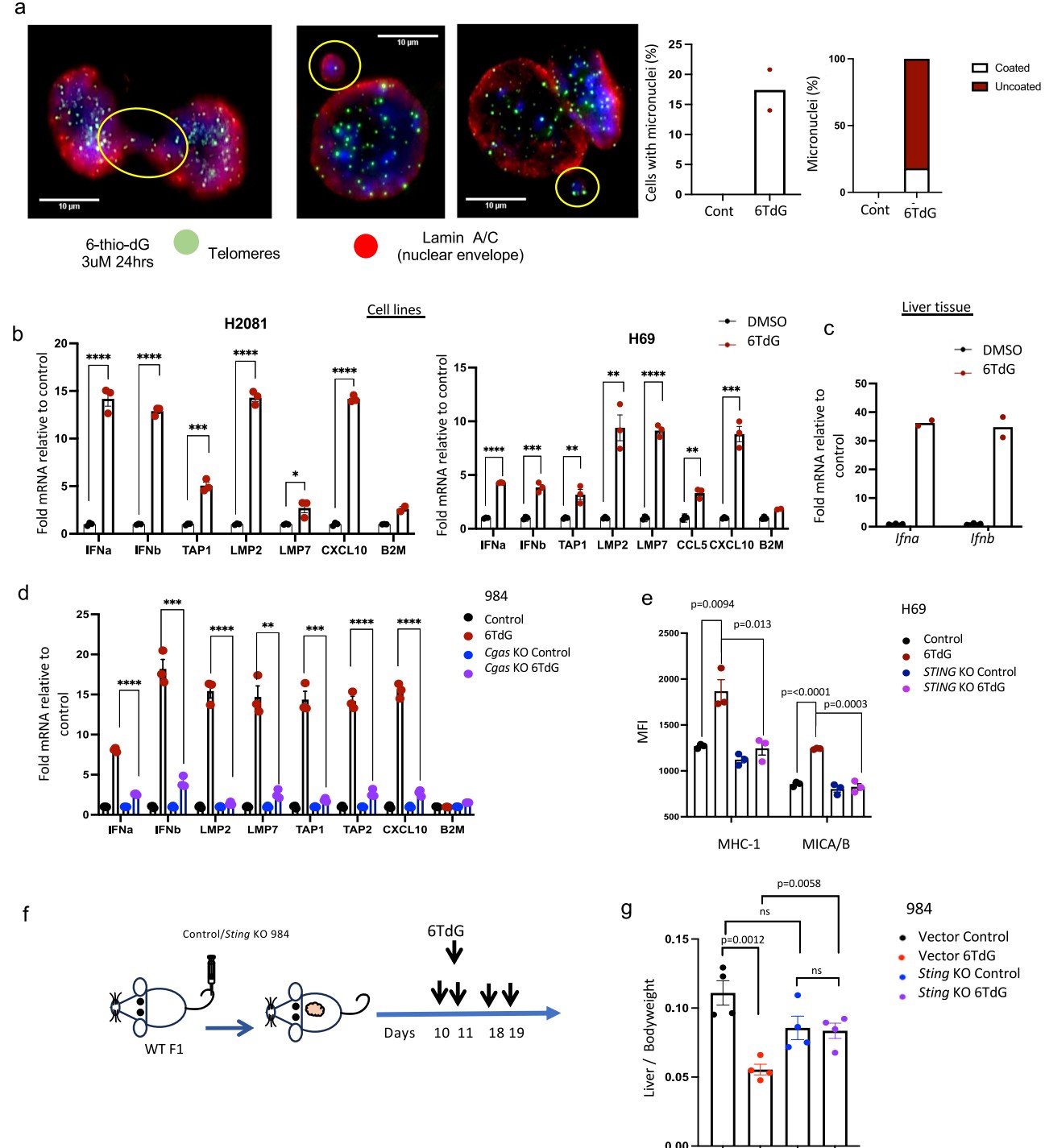

**Fig. 5 | 6TdG treatment activates cGAS-STING pathway. a** Representative immunofluorescence images of telomere (green) and lamin (red) stained 6TdG treated 984 cells showing micronuclei circled in yellow (left), quantification of micronuclei per sample in (middle) and distribution (% of total micronuclei) as coated or uncoated micronuclei in vehicle or 6TdG treated cells (right). $n = 2$ replicates per condition; experiment repeated twice. **b** qPCR data displaying fold change of mRNA for: *IFNa, IFNb, TAP1, LMP2, LMP7, CCL5, CXCL10, B2M* in vehicle vs 6TdG treated H69 and 2081 cells. All values were normalized to beta-actin before normalizing to vehicle treated samples. $n = 3$ for each group except 6TdG treated B2M condition ($n = 2$). Error bars represent mean + SEM. Two-tailed *T*-test was used to analyze each sample. (ns not significant, *$p < 0.05$, **$p < 0.01$, ***$p < 0.001$, ****$p < 0.0001$). **c** qPCR data displaying fold change of *Ifna* and *Ifnb* gene expression in 6TdG 984 liver tumors as compared to vehicle treated livers from Fig. 3. Samples were normalized to beta-actin before normalization to control. $N = 3$ for control

group and $n = 2$ for 6TdG treated group. **d** qPCR data displaying fold change of gene expression of *ifna, ifnb, Tap1, Lmp2, Lmp7, Ccl5, Cxcl10, B2m* in 984 C*gas* WT or KO cell treated with vehicle or 6TdG as compared to vehicle treated WT cells. $n = 3$ for each group. Error bars represent mean + SEM. Two-tailed *T*-test was used to analyze each sample. (ns not significant, *$p < 0.05$, **$p < 0.01$, ***$p < 0.001$, ****$p < 0.0001$). **e** MHC1 or MICA/B expression in H69 *STING* WT or KO cells treated with vehicle or 6TdG analyzed by flow cytometry. $n = 3$ for each group. Error bars represent mean + SEM. Multiple unpaired *T* tests were used for statistical analysis. **f** Schema for in vivo experiment with vector control or *Sting* KO 984 cells injected IV into WT mice and treated with vehicle or 6TdG. Mice were euthanized when vehicle controls reached the endpoint. **g** Liver/body weight ratios of mice from **f** are shown on the graph. $n = 4$ for each group. Error bars represent mean + SEM. Statistical analysis was completed by multiple unpaired *T* tests. Source data are provided as a Source Data file.

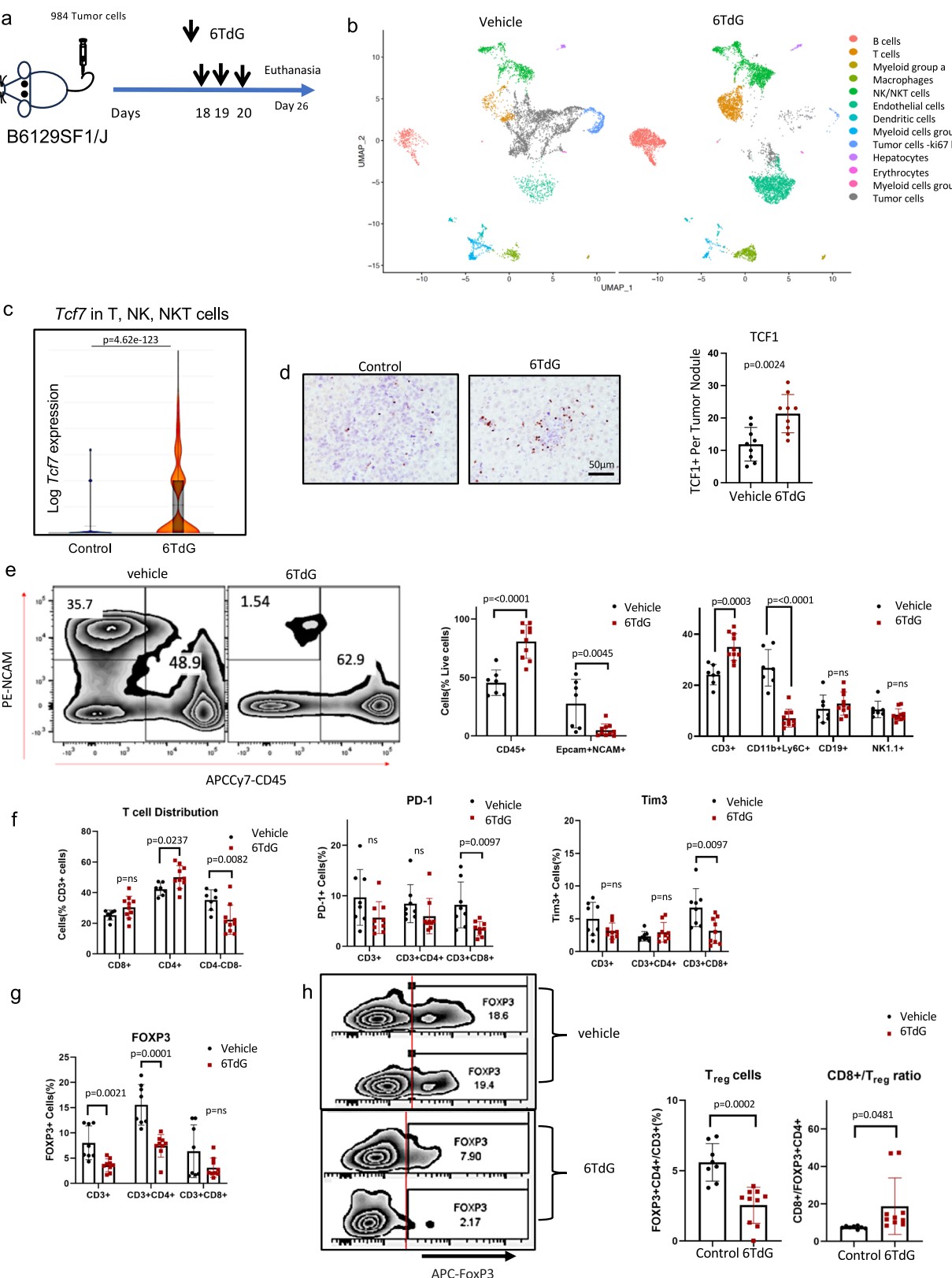

6TdG or IR vs combination groups, respectively) (Fig. 7b). Combination treatment also significantly increased the survival of treated mice with flank tumors as compared to controls in a separate experiment where mice were euthanized at the experiment endpoint (Fig. 7c, $p = 0.0015$). Combination treatment significantly increased DNA double stranded breaks, as compared to monotherapy with either of 6TdG (6TdG alone vs combination $p = 0.0007$) or IR (IR vs combination

$p = 0.014$), marked by gamma-H2AX on IHC (Fig. 7d). CD3 + T cell recruitment was also increased in all these treatment groups as compared to controls (Fig. S7a).

To determine whether 6TdG is effective in activating human T cells against human SCLC, we utilized a humanized mouse xenograft model. For this experiment, we used NSG-SGM3 mice, which allow superior engraftment of human immune cells due to constitutive

**Fig. 6 | 6TdG modulates tumor microenvironment. a** Schema for short-term in vivo experiment with 984 cells injected IV into WT mice, treated with 6TdG and sacrificed for downstream analysis. **b** Major cell populations from the single cell (10X) sequenced mouse liver tissues are shown in vehicle versus 6TdG treated groups as UMAP plots **c** *TCF7* expression among T, NK and NKT cells is quantified on the graph generated from single cell sequencing data. Maximum, minimum and mean value is 1.548, 0, and 0.075 for vehicle treatment group and 3.169, 0, and 0.512 for 6TdG treated groups respectively (*p* = 4.62e-123). Data was compared using ANOVA and *p* values adjusted with Benjamini–Hochberg correction for multiple tests. (Log2 of total UMI expression per cell 0.23 vs 0.51) Statistical test was negative binomial exact test in Cell ranger. 8–10 K cells from *n* = 1 mouse per condition were used. **d** Representative images of TCF1 IHC on the liver tumor tissue from the short-term treated mice as in panel a (left) and quantification of TCF1 positive cells (right). *n* = 3 mice per condition, 3 images from each mouse were used for analysis. Error bars represent mean + SEM and statistical analysis completed by unpaired *T* test. **e–g** Flow cytometry analysis of control (*n* = 7 or 8) and 6TdG –treated (*n* = 9 or 10) mouse livers after short term 6TdG treatment as in a. **e** Representative plot shows CD45+ or Ncam+ cells (left), quantification of CD45+ and NCAM+Epcam+ cells (middle) and major populations among immune cells (CD45+, right). Error bars represent mean + SD. Groups were compared by multiple unpaired *T* tests. **f** Distribution of T cell subsets (left) and T cell subsets expressing PD-1or Tim3 positive cells (%, right). Error bars represent mean + SD. Statistical analysis was completed by multiple unpaired *T* tests. **g** FOXP3 positivity of CD3,CD3 + CD4 + , or CD3 + CD8+ cells were quantified by flow cytometry. **h** Representative flow plots of T regulatory (Treg) cells (FoxP3 + CD4+ % among CD3 + T cells) in vehicle or 6TdG treated mice (left), FoxP3 + CD4+ are quantified in vehicle or 6TdG treated groups (middle) and the ratio of CD8 T cells to T regulatory cells in control versus treated groups (right). Error bars represent mean + SD. Statistical analysis was completed by multiple unpaired *T* tests. *N* = 8 control and 10 6TdG treated samples. Source data are provided as a Source Data file.

expression of cytokines SCF, GM-CSF, and IL-3[53]. We reconstituted human immunity in these mice with human cord blood hematopoietic stem cells (CD34 + ) for human tumor cell experiments (Fig. 7e). We pre-treated mice carrying H69 tumors with 6TdG before IR treatment and treated them with 5 Gy IR the next day (Fig. 7e). There was a significant additive effect with 6TdG and IR sequential combination, compared to either 6TdG alone (combination vs 6TdG alone *p* < 0.0001) or IR therapy alone (combination vs IR alone *p* = 0.0437) (Fig. 7f–h). Therapeutic benefit from 6TdG was lost in immunocompromised NSG mice (mice without cord blood reconstitution) implanted with H69 tumors while radiation therapy still had some therapeutic effect (Fig. 7g, h). All these data collectively suggest that 6TdG can sensitize SCLCs to IR and that this effect is dependent on an intact immune system, ultimately leading to prolonged survival in these preclinical models.

## Discussion

Induction of telomerase-mediated telomere dysfunction with the nucleoside analog 6TdG resulted in inhibition of tumor growth in preclinical models of SCLC. 6TdG treatment in vivo led to activation of the CD8 T cells and NK cells against tumor cells dependent on tumor cGAS/ STING signaling. This study demonstrates the immune-enhancing, metastasis reducing effects of the telomere-targeting agent, 6TdG, in several complementary in vitro and in vivo models of SCLC including syngeneic mouse, human xenograft, chemo-resistant, non-neuroendocrine variants of SCLC, and humanized xenograft models with patient derived SCLC cells.

Activation of DNA damage signaling can lead to inflammatory response against tumor through multiple cellular pathways. Upregulation of NK cell activating NKG2D ligands by DNA damage sensing proteins ATM and ATR, lead to NK cell activation against tumor cells[54,55]. Recent reports suggest that STING activating therapies can trigger the immune system against tumor cells and can enhance the effects of immunotherapy[56,57]. Activation of the STING pathway in SCLC through PARP inhibition was shown to synergize with immune therapy[15]. However, PARP inhibitor required daily treatments and is not effective as single agent in vivo. In contrast, 6TdG was effective as a monotherapy and lead to rapid anti-tumor response in mouse models of SCLC even with few doses. The combination of CIC depletion and STING activation may possibly account for potent activity of 6TdG as single agent. However, future studies can determine any other potential differences between these molecules.

We showed a significant increase in lymphocytes and specifically in T cells infiltrating into the tumor tissue while subsets of myeloid cells, and regulatory T cells were reduced with 6TdG treatment. Tumor infiltrating lymphocytes are shown to be associated with favorable clinical outcome in SCLC[58]. In contrast, myeloid derived suppressor cells (MDSCs) contribute to the immune resistance of tumors[59] and

regulatory T cells (FOXP3+) inhibit tumor-reactive T cells[60,61]. FOXP3 expression was shown to be associated with poor outcomes and resistance to immunotherapy in several tumors including NSCLC[60,62]. Depletion of regulatory T cells can enhance the effects of PD-L1 blockade[61,63]. In experimental systems, tumors in mice depleted of Tregs cells are more responsive to immunotherapies[64,65]. While these experimental Treg depleting strategies are not clinically feasible, a pharmacological option such as 6TdG treatment leading to anti-tumor immune microenvironment is potentially a viable alternative.

Tumor initiating cells within the SCLCs have higher telomerase activity and thus more sensitive than the bulk tumor to telomere targeting nucleoside analog 6TdG. The impact of 6TdG on CIC in SCLC is potentially of great importance for multiple reasons. One of the causes of acquired chemotherapy resistance in SCLC was shown to be increased intra-tumoral heterogeneity (ITH) and the presence of cancer initiating cells[66–70]. CICs are known to be associated with immunotherapy resistance as well, through several immune escape mechanisms including downregulation of MHC-I receptors and NKG2D ligands and downregulation of TLR-4 or upregulation of CD47 and PD-L1[50,71]. Additionally, several immunosuppressive mediators, including CCL2, CCL5, CSF1, and TGFβ, secreted from CICs, are shown to promote macrophage and Treg recruitment into tumor tissue, while suppressing T cell responses, contributing to immune resistance[72]. The depletion of CIC by 6TdG is also likely the potential reason of reduced metastasis with 6TdG treatment, despite the need to still identify the exact step of the metastasis inhibited by 6TdG, which needs to be explored in future studies. The therapeutic window available for 6TdG appears large since most normal tissues do not express telomerase and mice tolerated 6TdG treatment well. While activated T cells transiently activate telomerase, our findings are in keeping with the previous reports[36,37] and 6TdG treatment was well-tolerated by immunocompetent mice. Of course, human studies are needed to determine any potential effects on SCLC patients, particularly if it is given in combination with standard chemotherapy regimens.

## Methods
### Regulatory approvals
All mouse experiments in the manuscript were approved by institutional animal care and research committee (IACUC) at UTSW. The source of tumor material for PDX model generation was either core needle biopsy or circulating tumor cells. All tissue and blood samples from patients were collected per Mass General Hospital Institutional Review Board (IRB) approved protocols with written informed consent from the patients and in accordance with the Declaration of Helsinki. Human cord blood samples were obtained from UTSW/Parkland Hospital in compliance to the regulation and the use approval of human cord blood at UTSW medical center with consent from the patients.

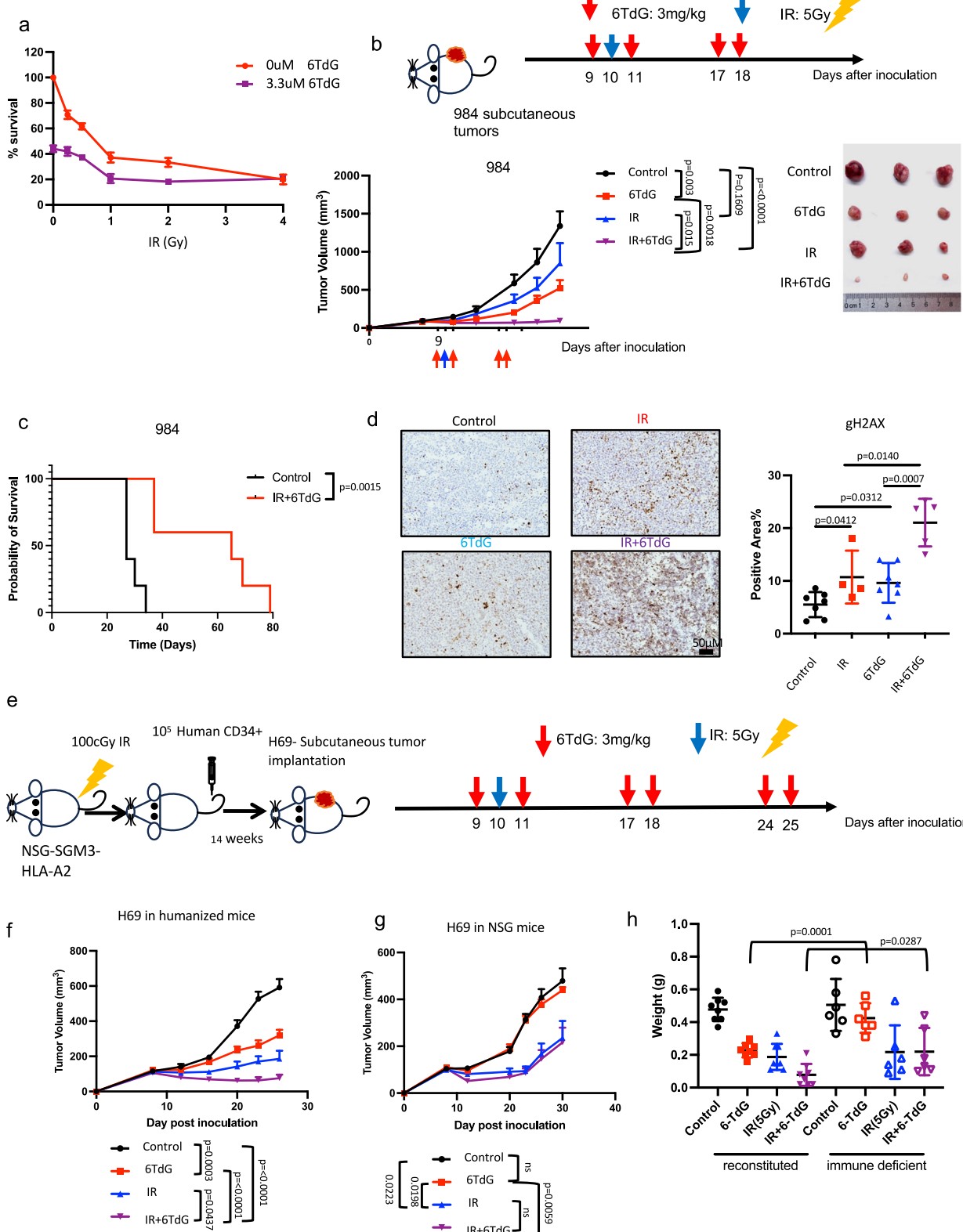

## Cell lines

Mouse lines 984 and 984 A were previously established murine-derived small cell lung cancer cell lines with Rb and p53 inactivation. They were generously shared by David McFadden at UT Southwestern. We reported RPP cell line previously[8], it was derived from an $Rb^{-/-}$; $p53^{-/-}$; $p130^{-/-}$ SCLC model. All these three cell lines were cultured in Dulbecco's modified Eagle medium (DMEM) with 10% FBS

and 1% antimycotic-antibiotic at 37 °C in a humidified atmosphere containing 5% $CO_2$.

Human small cell lung cancer cell lines (H510, H69, H841, H2081, H1048) were obtained from the Hamon Center for Therapeutic Oncology Research lines (University of Texas Southwestern Medical Center) where they are routinely fingerprinted, and mycoplasma tested. Human cells were cultured in RPMI with 10% FBS and 1%

**Fig. 7 | 6TdG potentiates radiation treatment. a** Dose response curves of H69 cells treated with combination of 6TdG (0 uM or 3.3 uM) and IR (0–4 Gy). $n = 5$ per group. % cell survival was quantified for each sample. **b** Schema for combination of IR (5 Gy) and 6TdG (3 mg/kg) treatment of subcutaneous 984 tumors. Mice received 6TdG on days 9,11, 17, and 18, and IR on day 10. Tumor volumes (mm³) for each treatment group (bottom left) and representative images of tumors (bottom right). $n = 7$ for each treatment group. Final tumor volumes were compared by multiple unpaired $T$ tests. Error bars represent mean + SEM. **c** A separate cohort of vehicle or combination of IR and 6TdG treated mice were evaluated for survival. $n = 5$ for each group. Mice were sacked when tumors reached 1500 mm³. Kaplan Meier analysis was performed to compare survival between the groups. Statistics performed using Log rank (Mantel–Cox) test. **d** Representative IHC images for gH2AX on the 984 subcutaneous allografts treated with vehicle, 6TdG, IR, or combination. gH2AX positive area (percentage among tumor tissue) was quantified

on the right. $n = 7$ for control, $n = 4$ in IR, $n = 7$ for 6-TdG and $n = 5$ for IR and 6TdG combination treatment groups. Error bars represent mean + SD. Multiple unpaired $T$ tests were used for statistical analysis. **e** Schema for establishing CD34+ cord blood cell humanized mice with H69 tumors and single agent or combination of IR and 6TdG treatment schedule is shown at the top. **f** Tumor growth curve for subcutaneous xenografts of human SCLC H69 treated with vehicle, IR, 6TdG or combination in reconstituted mice. $N = 8$ for each group. Error bars represent mean + SEM. Multiple unpaired $T$ tests were used for statistical analysis. **g** Tumor growth curve for subcutaneous xenografts of human SCLC H69 treated with vehicle, IR, 6TdG or combination in NSG-SGM3-HLA-A2 mice (non-reconstituted). $n = 6$ for each group. Error bars represent mean + SEM. Multiple unpaired $T$ tests were used for statistical analysis. **h** Comparison of final tumor weights of mice in **f** and **g**. Error bars represent mean + SD. Multiple unpaired $T$ tests were used for statistical analysis. Source data are provided as a Source Data file.

antimycotic-antibiotic at 37 °C in a humidified atmosphere containing 5% $CO_2$. Short-term patient derived xenograft (PDX) culture was obtained from serial dissociation of the PDX tumor tissue as detailed in the results.

### 3D colony formation assay
984 and H69 cells grow in suspension, 984 grows as tight spheres and H69 grows as disorganized clusters. $5 \times 10^3$ sorted cells were plated per well, in a 96 well plate and cell growth was observed for 5 days. Wells were photographed daily. Colony surface areas for each well were calculated using Image J (1.52a).

### 2D colony formation assay
Adherently growing mouse and human cell lines were used for these experiments. An equal number of cells were sparsely seeded in a 6 well plate. Cells were treated in one third dilutions of 6TdG (10 uM, 3 uM, 1 uM, 0.3 uM, 0.1 uM and control) in cell media for 7 days. At the end of treatment period, media was aspirated, cells were fixed with 10% formalin, and stained with crystal violet dye (1% crystal violet in 10% ethanol in PBS). The plates were washed with PBS, dried and scanned using BioRAD ChemiDoc. The number of colonies in each well was counted using Image J. Three repeats were performed for each cell line.

### 6TdG preparation
For in vitro treatments, the powder form of 6TdG (Metkinen Oy) was dissolved in DMSO/water (v/v = 1:1) to prepare 10 mmol/L stock solutions, aliquoted and kept frozen at −20 °C. Freeze-thaw cycles were avoided. For in vivo studies, 0.3 mg/mL stock solution was prepared by dissolving 6TdG powder in a 5%(v/v) DMSO solution diluted with PBS, aliquoted and stored at −20 °C. Freeze-thaw cycles were avoided. Mice were dosed with 3 mg/kg at the indicated schedules in respective figures.

### Cell viability measurements
An equal number of cells were counted and plated in each well of opaque-walled 96-well plates. Cells were treated with dilutions of 6TdG (10 uM, 3 uM, 1 uM, 0.3 uM, 0.1 uM, 0.03 uM, 0.01 uM, 0.003 uM, 0.001 uM and control) diluted in cell media for 5 days. For in vitro experiments a 1 mM stock of imetelstat was prepared by diluting the drug in PBS. Cisplatin stock solution was diluted in PBS for in vitro assays. This stock was further diluted in media for experiments. Three replicates were performed for each cell line unless otherwise indicated. A luminescent cell viability assay (CellTiter-Glo® Promega # G7570) was used to determine cell viability in each well. The manufacturer's protocol was used when incubating the cells and plates were protected from light during incubation process. The luminescence was measured using Tecan microplate reader. Survival curves were created and the IC50 for each cell line was calculated using Graph Prism.

### Senescence assay
Adherently growing murine and human SCLC cell lines 984 A, H841, and H510 were plated in complete media in a 6 well plate at 20% confluency. Cells were incubated with 0, 0.5, and 1uM of 6TdG for 48 h. Cells were washed to remove residual drug and incubated for 5 additional days. Cells were then stained with Senescence β-Galactosidase Staining Kit (Cell signaling technology, 9860). For each cell type and treatment dose three replicates were performed. At the end of the treatment period three images of each well taken using 10x lens, and images were analyzed using ImageJ software in a blinded manner. Senescent cells were counted as cells turning blue in color and quantified by number of cells per field of view.

### qPCR-Based telomerase activity assay
Telomerase Activity Quantification qPCR Assay Kit (TAQ)(ScienCell #8928) was used to quantitatively compare telomerase activity between CIC and non-CIC tumor cell populations. After sorting of L1CAM negative and L1CAM positive or CD133 high and CD133 low populations separated via FACS, cells were lysed and prepared according to manufacturer's protocol. The cell lysis buffer is a mild lysis buffer that enables the release of telomerase in the native state. The telomere primer set (TPS) recognizes and amplifies newly synthesized telomere sequences in the assay. Quantification cycle value difference between the two samples (ΔCq, Telomere primer set (TPS)) according to manufacturer's instructions for each cell population and was compared to pooled/unsorted tumor cell population. ΔCq (TPS) was calculated as ΔCq (TPS) = Cq (TPS, sample 2)−Cq (TPS, sample 1). Three replicates were performed for each cell line.

### Telomere dysfunction induced foci (TIF) and micronuclei assay
H841 cells were seeded, treated with DMSO or 5 μM 6TdG, respectively, for 48 h and slides were rinsed in PBS for 15 min on a shaking platform. Slides were fixed in 4% formaldehyde (Thermo Fisher) for 10 min on ice and then washed in PBS for 5 min, two times. Subsequently, slides were permeabilized with 0.5% Triton X-100 for 10 min on ice and then blocked with BSA/PBS for 30 min at room temperature (RT). Anti-mouse primary antibody γH2AX (Millipore, #05636) was diluted 1:200 in blocking solution (TIFs) or lamin A/C (Sana Cruz, E-1) (micronuclei) and cells incubated in a humid chamber at 4 °C ON. Following washes with PBS, cells were incubated with AlexaFluor 488 conjugated goat anti-mouse (TIFs) or Alexaflour 568 conjugated goat anti-mouse antibody (Invitrogen), for 45 min at RT. After PBS 1X washes, cells were fixed in 4% formaldehyde in PBS for 20 min at RT. The slides were sequentially dehydrated with 70%, 90%, 100% ethanol and subsequently denatured for 3 min at 80 °C with 20 μ of hybridization mixture contained 70% deionized formamide, 1 M Tris pH 7.2, 8.56% buffer MgCl2, 5% maleic blocking reagent, and 25 μg/mL Cy3-conjugated PNA Tel-C (CCCTAA)3 probe (Panagene, #F1002, South Korea) and incubated overnight at 4 °C in a humidified chamber. Slides were washed

two times for 15 min in a wash solution containing 70% formamide, 10 mM Tris pH 7.2, 0.1% BSA, and washed three times for 5 min in a solution containing 0.1 M Tris pH 7.5, 0.15 M NaCl, and 0.08% Tween-20. The slides were dehydrated by ethanol series, air-dried, and counterstained with Vectashield/DAPI (Vector Laboratories, Burlingame, CA). Images were captured at 63X magnification with an Axio Imager Z2 (Carl Zeiss) equipped with an automatic capture system (Metafer, Metasystems) and analyzed with ISIS software (Metasystems). One hundred nuclei were counted for TIFs assay, and one thousand nuclei were counted for micronuclei analysis.

## Collection of chromosome spreads
Cells were treated with 5 µM colchicine (Sigma-Aldrich) and incubated for 5 h at 37 °C to enrich the fraction of mitotic cells. Successively, cells were collected and re-suspended in 75 mM KCl hypotonic solution (Sigma-Aldrich) for 20 min at 37 °C, followed by fixation in Carnoy solution (3:1 v/v Methanol/Acetic Acid) (Fisher Scientific, Hampton, NH, USA). Finally, cells were dropped onto slides and air dried.

## Telomeric qualitative fluorescent in situ hybridization
Cells were fixed and seeded as described in the previous section (Collection of chromosome spreads). After 48 h, slides were rinsed in 1X PBS for 15 min and fixed in formaldehyde 4% (Sigma-Aldrich) in PBS for 2 min. Successively, slides were rinsed in 1X PBS and incubated in acidified pepsin solution for 10 min at 37 °C, then washed in 1X PBS. The slides were fixed again in formaldehyde 4% in PBS for 2 minutes, washed in 1X PBS two times and dehydrated in 70%, 90%, and 100% ethanol (Fisher Scientific) for 5 min each and air-dried. The slides were denatured for 3 min at 80 °C with 20 µL of hybridization mixture containing 70% deionized formamide, 1 M Tris pH 7.2, 8.56% buffer MgCl2, 5% maleic blocking reagent, and 25 µg/mL Alexa 488-conjugated PNA Tel-G (TTAGGG) probe (PNA Bio, US), Cy3 conjugated CENPB (ATTCGTTGGAAACGGGA, (PNA Bio) and incubated overnight at 4 °C in a humid chamber. Slides were washed two times for 15 min in wash solution containing 70% formamide, 10 mM Tris pH 7.2, 0.1% BSA, and washed three times for 5 minutes in a solution containing 0.1 M Tris pH 7.5, 0.15 M NaCl, and 0.08% Tween-20. The slides were dehydrated by ethanol series, air-dried, and counterstained with Vectashield/DAPI (Vector Laboratories, Burlingame, CA). Images were captured at 63X magnification with an Axio Imager Z2 (Carl Zeiss) equipped with an automatic capture system (Metafer, Metasystems) and analyzed with ISIS software (Metasystems).

## Co-culture experiments
Tumor cells were treated with vehicle or 1uM 6TdG for 72 h. CD3+ cells were isolated (using Biolegend #480021), CD4+ isolated (using Biolegend #480009), or whole human PBMCs were co-cultured with 6TdG pre-treated SCLC lines at 10:1 ratio of immune cells to tumor cells. For CD3 and PBMC co-cultures, T cells were activated with CD3 (1 ug/mL) and CD28 (5 ug/mL) antibody. Cells were cultured together for 48 h and harvested for flow cytometry; media was harvested for ELISA analysis.

## Flow cytometry
Cell lines were stained with a fixable live/dead cell stain (Fisher Cat# 50-112-1528) for 8 min at room temperature, followed by staining with fluorophore-conjugated antibodies for 20 min on ice and in the dark. Cells were washed twice with a FACS buffer (2% FBS in PBS) after every staining step and analyzed using a BD FACS Canto machine or Cytek Northern Lights. Compensation was completed if samples were stained with multiple antibodies. The data were analyzed using the FlowJo software (v10.7).

Mouse tissues were minced and treated with DNase and collagenase (100 units/ml Collagenase IV Fisher Cat# 17104019, 10 µg/ml DNase I Sigma Cat# DN25, 10% heat inactivated FBS in RPMI) for 1 h at 37 degrees to dissociate the cells and then passed through a cell strainer (70 µm) to create a single-cell suspension. Red blood cells (RBC) were removed from the sample using RBC lysis buffer (Fisher Cat# A1049201). These cells were then stained with a live/dead cell stain, followed by staining with fluorophore-conjugated antibodies as detailed above, following FC blocking with CD16/32 antibody (Biolegend, Cat# 101320) for 20 min on ice. The samples were washed twice after every step. To analyze intracellular markers, the samples were stained with surface markers and then permeabilized at 4 °C overnight and incubated with fluorophore-conjugated antibodies recognizing intracellular markers. The manufacturer's instructions were followed for the intracellular staining (eBioscience Foxp3/Transcription Factor Staining Buffer Set, Thermofisher, Cat# 00-5523-00). Detailed antibody information is provided in Table S1.

Flow cytometry analysis was performed using BDFACS Canto, and flow data was analyzed using FlowJo (v10.7). Compensation was performed for multi-color stains. Example gating strategy is provided in Fig. S8.

## ELISA
Media from both CD3 + PMBC or whole PBMC co-cultures were collected for both IFNG and Granzyme B ELISA. ELISA's were completed using manufacturer's instructions for human IFNG (Biolegend human IFN-γ ELISA MAX kit, cat # 430107) and Human Granzyme B (Biolegend, Cat # 439207).

## CBC counts and marrow cultures
Blood samples were collected through terminal cardiac puncture from long-term vehicle or 6TdG treated mice and clotting prevented by adding 0.5 M EDTA. The whole blood was analyzed with Element HT5 CBC differential machine. Data was reported as either counts or percentages of whole CBC populations. Bone marrows were isolated from femurs and dissociated. Cells were plated as per the instructions of MethoCult media (Stem cell Technologies), and colonies were quantified 7 days after plating.

## RNA extraction and qRT-PCR
RNA was extracted by Qiagen RNAeasy kit. 1 µg of total RNA was reverse transcribed to cDNA by Applied Biosystems TaqMan High-Capacity RNA-to-cDNA Kit (Fisher, # 43-874-06) in 20 µl volume. 1 µl cDNA product was used for qPCR in by Sybr green master mix (Biorad, 1725270) or Taqman real-time prc master mix (with Taqman probes). Amplification was assessed by QuantStudio 3 Real-Time PCR System. Primers for human genes: TAP1F: GGGGACAGCTGCTGTTGGAT, TAP1R: AGTACACACGGTTTCCGGATCAAT; B2MF: CCACTGAAAAAGATGAGTATGCCT, B2MR: CCAATCCAAATGCGGCATCTTCA, CCL5F: TCATTGCTACTGCCCTCTGC, CCL5R: TACTCCTTGATGTGGGCACG, CCL10F: GGTGAGAAGAGATGTCTGAATCC, CCL10R: GTCCATCCTTGGAAGCACTGCA, LMP2F: CGAGAGGACTTGTCTGCACATC, LMP2R: CACCAATGGCAAAAGGCTGTCG, LMP7F: CCTTACCTGCTTGGCACCATGT, LMP7R: TTGGAGGCTGCCGACACTGAAA, ATCBF:TGACCCAGATCATGTTTGAGA TGACCCAGATCATGTTTGAGA, ATCBR: TACGGCCAGAGGCGTACAGC, Taqman IFNA1: Hs04189288_g1 and IFNB1: Hs01077958_s1, ACTB: Hs01060665_g1. Mouse primers: ifna1F: GGATGTGACCTTCCTCAGACTC, ifna1R: ACCTTCTCCTGCGGGAATCCAA, ifnbF: GCCTTTGCCATCCAAGAGATGC, ifnbR: ACACTGTCTGCTGGTGGAGTTC, Tap1F: GCTGTTCAGGTCCTGCTCTC, Tap1R: CACTGAGTGGAGAGCAAGGAG, B2mF: CTGGTGCTTGTCTCACTGACC, B2mR: GACCAGTCCTTGCTGAAGGAC, Ccl5F: CCTGCTGCTTTGCCTACCTCTC, Ccl5R: ACACACTTGGCGGTTCCTTCGA, Cxcl10F: ATCATCCCTGCGAGCCTATCCT, Cxcl10R: GACCTTTTTTGGCTAAACGCTTTC, Lmp2F: CCTCTGCACCAGCACATCTTC, Lmp2R: CCTCTGCACCAGCACATCTTC, Lmp7F: GGACCTCAGTCCTGAAGAGG, Lmp7R: CAACCGTCTTCCTTCATGTGG, ActbF: GGTCCACACCCGCCACCAG, ActbR: CACATGCCGGAGCCGTTGTC.

## Western blotting

Cells were lysed in RIPA buffer (Cell signaling technology, 9806) with protease/phosphatase inhibitor cocktail (Cell signaling technology, 5872) and agitated at 4 °C for 30 min, followed by centrifugation at 2000 g for 20 min. Supernatants were collected and protein concentration was calculated by the Pierce BCA Protein Assay Kit (ThermoFisher Cat # 23225) according to manufacturer's protocol. Equal amounts of protein are diluted in SDS running buffer and reducing reagent (ThermoFisher #NP0002 and #NP0004) and subjected to electrophoresis and immunoblotting has been performed according to manufacturer's protocol (BioRad MiniProtein Electrophoresis cell #1658004 and BioRAD Criterion Blotter #1704070). The immunoblot membranes were blocked in 5% milk in TBST 1 h at room temperature, washed (TBST, 10 min) and incubated with primary antibodies diluted in 5% milk or BSA (for phospho antibodies) in overnight at 4 °C followed by a rigorous wash (5 min x4). Secondary antibody of appropriate species was diluted in 5% milk or BSA in TBST at 1:5000 dilution and incubated at room temperature for 1 h. After washing, membranes were incubated with chemiluminescence substrate (ThermoFisher # 34580) and imaged using (BioRAD ChemiDoc). Antibodies are listed in Table S1.

## In vivo tumor models

According to our IACUC approved protocol maximum allowed tumor size is 2 cm in the largest dimension. 984 cells (WT, plasmid control, *Tert* KO or *Sting* KO) were implanted either intravenously (IV) or subcutaneously into wildtype B6129SF1/J mice. Plasmid is plenti-CrisprV2. $1 \times 10^6$ 984 cells were used for subcutaneous models on WT mice, $2 \times 10^5$ cells were used in IV experiments except for L1CAM high/low sorted cell experiments. $1 \times 10^5$ sorted L1CAM high or low cells were injected by IV. 984 cells have higher tropism for liver tissue and form macroscopic liver tumors when injected intravenously via tail vein into WT mice, with micro-metastasis to lung tissue.

RPP cells were implanted into nu/nu mice (Jax # 007850). $5 \times 10^6$ RPP cells were used in subcutaneous implantation experiments in nude mice. Width (shorter dimension) and length (longer dimension) of tumors were measured by digital caliper and volumes were calculated by this formula: width (mm) x width (mm) x length (mm)/2. Both male and female mice were included in the study. At the endpoint of the experiments, mice were euthanized and tumor bearing livers or flank tumors were collected. One lobe of the livers or part of flank tumors were fixed in formalin to be embedded in paraffin for H&E and IHC analysis, and the rest of the liver/tumor tissues were used for flow cytometry analysis.

## Treatment doses and schedules

6TdG was dosed at 3 mg/kg (in 5% DMSO) at the indicated days in the figures.

Cisplatin was administered 2 mg/kg once a week and etoposide 10 mg/kg three times a week. The drugs were diluted in sterile saline.

1 mM imetelstat stock solution was prepared by dissolving 10 mg GRN163L in 2 ml PBS. Imetelstat was dosed at 30 mg/kg (in PBS) twice a week.

Cell depleting antibodies: NK cells were depleted using anti-NK1.1 (BioXcell, anti-mouse NK1.1 clone PK136) or anti-CD8 antibody was depleted using anti-mouse CD8a (BioXcell, anti-mouse CD8a, clone 2.43). Antibodies were administered intraperitoneally to the mice at 200ug/dose at the indicated schedules.

Local or whole-body radiation treatments: Radiation treatments were performed on SAARP from Xstrahl Inc. by the Preclinical radiation core facility. For tumor treatments, a single fraction of stereotactic radiation at the indicated dose was given with image guidance using the SARRP system (Xstrahl Inc).

## Humanized mouse model

Transgenic mice strain NSG-HLA-A2 (JAX #014570) and NSG-SGM3 (Jax #013062) were crossed to generate NSG-HLA-A2-SGM3. NSG-HLA-A2-SGM3 mice were whole-body irradiated at 4 weeks at 100 cGy. Donor human umbilical cord blood was first balanced with equal volume of PBS and then 35 ml of balanced blood was added to 15 ml of Lymphoprep Density gradient medium (Stem Cell technologies, Cat#7861) to isolate mononuclear cells. Human CD34+ hematopoietic stem cells (HSCs) were isolated from processed mononuclear cells using EasySep Human Cord Blood CD34 Positive Selection Kit (Stemcell, Cat#17896). $1 \times 10^5$ human CD34+ HSCs were IV injected into irradiated NSG-HLA-A2-SGM3 mice 24 h after radiation. Mice were kept on Bactrim water for 4 weeks after the procedure to prevent infection. Human CD45+ reconstitution was confirmed at 12 weeks post HSC inoculation in peripheral blood by flow cytometry. Reconstituted mice were then subcutaneously inoculated with $5 \times 10^6$ H69 cells. Both male and female mice were included in all studies. Tumors were measured and volumes were quantified as detailed above. $5 \times 10^6$ H69 were also inoculated into un-reconstituted NSG-HLA-A2-SGM3 mice.

## Immunohistochemistry

Slides were first deparaffinized and hydrated. Then they were incubated in gently boiling 35 mM sodium citrate solution for 15 min for antigen retrieval. Slides were cooled down to RT and blocked with 3% H2O2 for 30 min. The slides were then washed and blocked with 1% BSA in PBS for 1 min at RT. Primary antibodies γH2A.X (Cell signaling Technology, # 9718), CD8a, (CST,# 13647), Cleaved caspase-3, (CST, #9661), TCF1/TCF7 (CST, #2203), Ki67, (Thermo Fisher, # MA5-14520) were diluted in PBS with 2% BSA and incubated with the slides at RT for 1 h. Slides were then washed with 4xTBST for 5 min. Goat Anti-Rabbit secondary antibodies (ImmPRESS HRP Goat Anti-Rabbit IgG Polymer Detection Kit, Vector Lab, # MP-7451) was used as secondary staining. The slides were incubated at RT for 1 h with secondary antibodies. Slides were washed with TBST for 5 min 4 times. Substrate (ImmPACT DAB Substrate, Peroxidase, Vector Lab, # SK-4105) was applied on the slides and color formation was observed. Slides were counterstained with hematoxylin, washed, air dried and mounted. Slides were covered with coverslips and scanned with a Zeiss AX10 microscope with CoolCube metasystem. Tumor volumes on H&E slides are analyzed using QuPath (v0.5.0.). IHC stained slides (except for ki67) were quantified using Image J (version 1.52a) and IHC analysis toolbox plugin (NIH). Ki67 staining was quantified using QuPath (v0.5.0) software.

## Single cell RNA sequencing and analysis

Mice were treated according to the short-term treatment schedule and mouse livers were processed for single cell sequencing. Single cell RNA sequencing was completed at Microbiome Research Laboratory at UT Southwestern Medical Center. About 8–10 K single cells with viability score of >70% were input in each library. The 10x Genomics Chromium controller instrument was used for Gel Bead- in Emulsion (GEMs) preparation. Chromium Next GEM Single Cell 3' Kit v3.1 (PC-1000269), ChromiumNext GEM Chip G Single Cell Kit (PC-1000127) and Dual Index Kit TT Set A Kit (PC-1000215) were used for Single-Cell Library Preparation. cDNA and final barcoded sequencing libraries were generated as per the manufacturer's specifications and their quality and concentration was assessed using the Bioanalyzer 2100 and qPCR, respectively. Quality pass single cell libraries were sequenced on NextSeq550 sequencer using the paired end 75 bp sequencing kit. About 20–30 K sequencing reads were generated per cell. UMI counts for each cellular barcode were quantified and used to estimate the number of cells successfully captured and sequenced.

Cell Ranger Single-cell Software Suit (v 5.0.0) was used for sample demultiplexing and single-cell 3' gene counting. SoupX R package (constantAmateur/SoupX: R package to quantify and remove cell free

mRNAs from droplet based scRNA-seq data (github.com)) was used for ambient RNA reads cleaning. Doublets were identified using Scrublet (Scrublet: Computational Identification of Cell Doublets in Single-Cell Transcriptomic Data−ScienceDirect) and removed from downstream analysis. We further filtered cells with more than 15% of the transcripts coming from mitochondrial genes and cells with Count_RNA > 10000 or <100 genes detected, allowing us to exclude low quality cells that experience cell death process. In total, we retained 7121 & 5433 cells for Treated & Cont Condition, respectively.

To account for potential batch effects, we performed integration with SCTtranform-normalized datasets. The integration was performed using Seurat R package (v3), in brief, the Cont & Treat samples were normalized using SCTransform function in the Seurat R package and with vars.to.regress parameter set to "percent.mt". Features for downstream integration were identified using SelectIntegrationFeatures function and was used as input for PrepSCTIntegration function. Integration anchors were identified using FindIntegrationAnchors function with normalization.method set to "SCT" and anchor.features set to above identified integration anchors. The final integration was performed using IntegrateData function with normalization.method set to "SCT" and anchorset set to previous identified anchors. Finally, we performed dimension reduction using function RunPCA & RunUMAP with dims set to 30. Unsupervised clustering was performed using FindClusters function with resolution set to 0.1. The cell cluster types were determined based on the top marker gene expression and literature[73].

### Statistical analysis
Data analysis was performed with GraphPad prism except for expression analysis. Specific test used for each experiment is indicated within the figure legends.

### Reporting summary
Further information on research design is available in the Nature Portfolio Reporting Summary linked to this article.

## Data availability
The scRNA-seq data was deposited to GEO with accession number GSE225018. The remaining data are available within the Article, Supplementary Information or Source Data file. Source data are provided with this paper.

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

## Acknowledgements

We thank Drs Jane Johnson and David McFadden for mouse SCLC cells; Dr Jaehyup Kim for hematopathology expertise; Dr Qing Deng for proofreading; late Dr Benjamin Chen for help with radiation experiments; and Daryl Harmon and Jade Lee for administrative assistance. UTSW Tissue Management Shared resource for tumor tissue embedding and sectioning (P30CA142543) and Whole Brain Microscopy Facility for slide scanning (RRID: SCR_017949). EAA is a Cancer Prevention and Research Institute of Texas (CPRIT) Scholar in Cancer Research. This work was supported by CPRIT Scholar Award RR160080, NIH-R01CA276058, Department of Defense (DOD) W81XWH-21-1-0856, Welch Foundation grant (1975-20190330), A Breath of Hope Lung Foundation Fellowship Award (ABOHLF 2020), American Cancer Society Research Scholar Award RSG-22-051-01-IBCD, and Mary Kay Ash Foundation grants to E.A.A., NIH 5P50CA070907 to J.D.M., J.W.S. and E.A.A. R.R.K. was supported by 5T32CA124334. J.D.M. is also supported by U01CA213338. Preclinical Radiation Core Facility is supported by CPRIT (RP180770). J.W.S. holds the Southland Financial Corporation Distinguished Chair in Geriatrics Research.

## Author contributions

B.E., M.Z. and E.A.A. conceived the study. B.E., M.Z., R.R.K., H.H. and M.B. performed the experiments. S.S. conducted TIF, micronuclei analysis, chromosome analysis and I.M. conducted PDX experiment, P.R. performed single cell sequencing and initial analysis. K.C. and L.X. conducted computational and data analyses. J.W.S. provided resources and edited draft. B.D., V.S. and J.D.M. provided resources. B.E. and E.A.A. wrote the original manuscript with input from authors. R.R.K. and E.A.A. revised the manuscript.

## Competing interests

I.M., S.S. and J.W.S. are named inventors on patents licensed to MAIA (16/450,430; 62,636,775; 62/646,820; 16/304,538; 17/200,539; 63/388,688). J.W.S. is on the SAB of MAIA Biotechnology. I.M. is currently at Maia Biotechnology Inc, Chicago, Illinois, USA. J.D.M. receives licensing fees from the NCI and UT Southwestern to distribute cell lines. B.J.D. has consulting agreements with Astra Zeneca, Sonata Therapeutics and Dialectic Therapeutics. Other authors declare no relevant conflicts of interests.

## Additional information

[1]Department of Pathology, University of Texas Southwestern Medical Center, Dallas, TX, USA. [2]Simmons Comprehensive Cancer Center, Dallas, TX, USA. [3]Department of Cell Biology, University of Texas Southwestern Medical Center, Dallas, TX, USA. [4]Quantitative Biomedical Research Center, Department of Population & Data Sciences, University of Texas Southwestern Medical Center, Dallas, TX, USA. [5]Hamon Center for Therapeutic Oncology, University of Texas Southwestern Medical Center, Dallas, TX, USA. [6]Department of Immunology and Microbiome Research Laboratory University of Texas Southwestern, Dallas, TX, USA. [7]Department of Pharmacology, University of Texas Southwestern Medical Center, Dallas, TX, USA. [8]Department of Internal Medicine, University of Texas Southwestern Medical Center, Dallas TX, Medical Center, Dallas, TX, USA. [9]Department of Pediatrics University of Texas Southwestern Medical Center, Dallas, TX, USA. [10]These authors contributed equally: Buse Eglenen-Polat, Ryan R. Kowash. ✉e-mail: esra.akbay@utsouthwestern.edu

