## [Peer Review File · Nature Communications]

A telomere-targeting drug depletes cancer initiating cells and promotes anti-tumor immunity in small cell lung cancerEditorial Note: Parts of this Peer Review File have been redacted as indicated to remove third-party material where no permission to publish could be obtained.

REVIEWER COMMENTS

Reviewer #1 (Remarks to the Author): with expertise in lung cancer immunology, SCLC, STING

In this manuscript, Eglenen-Polat and Colleagues describe a new potential therapeutic approach for small cell lung cancer (SCLC) by treating with a nucleoside analog, 6-Thio-2'-deoxyguanosine 6TdG. They find that inducing telomere dysfunction can efficiently inhibit primary SCLC tumor growth and metastasis from the primary tumors, preferentially depleting cancer initiating cells with higher telomerase activity. The authors suggest that tumor control is mediated by NK and the T cells and observe a synergistic effect of telomere damage therapy with radiotherapy in SCLC. The authors suggest that this anti-tumor effect is tumor-STING dependent, however, the data does not fully support this conclusion with strong evidence. While the concept has several implications for small cell lung cancer patients and is potentially interesting for the readers of this journal, additional work is needed to substantiate their claims, particularly with respect to the data presented in Fig. 5.

Major points:

Figure 5A, C: The authors showed that 6TdG treatment for three days induces surface MHC-I expression by flow cytometry in mouse SCLC lines in a dose dependent manner and in one human cell line (H69). Have the authors examined induction of surface MHC-I expression in additional SCLC lines to recapitulate the same effect?

Figure 5B: Regarding studies of NK cell activating ligands-NKG2DL in mouse SCLC cells - have the authors examined activating and inhibitory ligands on the surface of several SCLC human lines?

Figure 5C: Could the authors explain which NKG2D ligands that they were screening for in

H69? MICA/B?

Figure 5D: The methods/conditions for these experiments are unclear, and the figure legend does not explain if intracellular flow staining was utilized. PBMCs cells were used and then gated on CD3+ cells. It would be nice to repeat this experiment using isolated CD3+ T cells for co-culture, also measuring secreted cytokines such as IFN γ and Granzyme B ELISA to support their claims.

Figure 5F: IFN β induction by 6T-dG seems to fall into low levels of detectability. Also, the authors do not mention which mouse cell line was tested. Could the authors screen additional human cell lines to recapitulate this effect in human models?

Figure 5H: To determine the contribution of STING signaling in mediating 6TdG effects, the authors showed a loss of function experiment in one human and mouse cell line. The authors claimed in their abstract that mechanistically “Mechanistically, 6TdG depleted induced type-I interferon signaling by activating tumor STING”. However, they have not directly examined activation of the cGAS-STING pathway and TBK1-IRF3 signaling. Firstly, the authors should validate their claims using additional lines, and mechanistically prove STING activation by immunoblotting for canonical STING pathway proteins (eg pTBK1, pIRF3 and/or pSTAT1). Moreover, STING activation is known to cause chemokine production (such as CXCL10, CCL5) which will enhance immune cell infiltration, however none of these chemokines were screened.

Minor points:

Figure 3H, statistical analysis is missing in the first panel.

In the results section, a reference about the first line of SCLC therapy is missing.

Reference to Fig. 5G missing in the manuscript.

The authors should consider moving the introduction on cGAS-STING pathway from results into the introduction section.

Reviewer #2 (Remarks to the Author): with expertise in cancer, telomeres/telomerases

In this study, Eglenen-Polat et al. investigate the chemotherapeutic mechanism of 6-thio-2'-deoxyguanosine (6TdG) as an anti-telomerase activity strategy in small cell lung cancer (SCLC) models, both in vitro and in vivo. They demonstrate that 6TdG treatment exhibits significant anti-tumor activity by inducing DNA damage at telomeres, also referred to as telomere dysfunction-induced foci (TIFs), in cancer initiating cells (CICs) that predominantly depend on telomerase activity for proliferation.

The authors investigate the anti-tumor activity and reveal that TIFs subsequently activate the STING pathway, generating a favorable immune response within the tumor microenvironment and leading to the recruitment of T cells and natural killer (NK) cells. Additionally, they extend their findings by illustrating that 6TdG treatment, when combined with ionizing radiation (IR), could serve as an effective alternative for SCLC treatment by exhibiting a synergistic effect in vivo.

These results hold relevance for the field and offer a new direction for targeting in specific cancer subtypes, yet I am not convinced that this is through a telomere deprotection pathway. The only panel that speaks to this is 1E. The telomere connection has to be strengthened or removed. The presentation and quality of the data is not up to the standard for Nature Communications.

Major concerns:

These results hold relevance for the field and offer a new direction for targeting in specific cancer subtypes, yet I am not convinced that this is through a telomere deprotection pathway. The only panel that speaks to this is 1E. The telomere connection has to be strengthened or removed. The presentation and quality of the data is not up to the standard for Nature Communications.

In many cases the quality and resolution of the figures presented is poor and does not allow interpretation of the data. This is particularly relevant for IHCs images and

immunofluorescence. H&E is overstained, pictures are out of focus, etc. Size bars are generally missing. Figure 1G is not convincing that this reflected telomere colocalization and qualifies as a TIF.

The authors should specify how statistics have been used in each figure panel. In particular the statistics used behind the p values indicated in the figures is not clear. In addition the authors should indicate how many times each experiment has been performed.

The majority of the graphs are difficult to read as in most cases the authors do not specify what the dots in the graph actually represent. We suggest the authors clearly state what dots represent in each graph.

Point by point comments:

Figure 1E-F, please indicate how many biological replicates have been used for this panel. In addition please specify in more detail the statistics used for this figures.

Figure 1G, the quality of the images presented is low and prevents a correct interpretation of the results. In addition it is not indicated how many times this staining has been performed.

Figure 2B, please indicate what the dots in the bar plots represent. Are these 3 independent experiments?

Figure 4A-C, the effective depletion of CD8+ and NK cells is not shown. Please provide data showing that this depletion treatment was effective.

Figure 5B, please indicate clearly what the dots are representing.

Page 17: "Therapeutic efficacy of 6TdG was lost: there was no significant difference between metastatic tumor burden of 6TdG treated or control STING-KO tumors, suggesting that in this SCLC model system the therapeutic effects of 6TdG is tumor STING-dependent, an important difference from other immunogenic models and highlights tumor STING as a vulnerability of SCLC". This paragraph misses a reference to a figure. Is this referred to Fig. 5H?

Fig. 6E, the FACS plot graph presented is of poor quality and difficult to interpret. Please provide a higher quality plot for this data.

Fig. 6G, the FACS plot graphs presented are barely interpretable. Please reorganize these panels and provide a higher quality presentation of this data.

Fig. 3K, possibly erroneous figure call. It looks like this is actually referred to Fig. 6H.

Fig. 7A, this data are better represented as dotted lines as in previous panels, please modify it to dotted lines representation.

Reviewer #3 (Remarks to the Author): with expertise in SCLC, tumor initiating cells, STING

This study by Eglenen-Polat and Zhu et al. depicts the therapeutic potential of telomere-targeting agent, 6TdG in small cell lung cancer (SCLC). The authors show that 6TdG treatment causes cell death in vitro and increased DNA damage both in vitro and in vivo. Authors also show that there is a decrease in tumor burden in 6TdG compared to Cisplatin-etoposide treated mice, however the difference is modest and the method of statistical analysis performed is not disclosed. When SCLC populations are enriched for cancer initiation cells (CICs), 6TdG treatment shows increased cell death in vitro and decreased tumor burden in vivo. Authors conclude that 6TdG treatment target CICs in vivo. Authors investigate a previously described observation that 6TdG treatment induces an immune response in vivo, by depleting CD8 or NK cells and show a loss of 6TdG efficacy in regards of tumor burden. To further explore this observation, authors show that there is an increase in MHC-I expression after 6TdG treatment in vitro and an increase of a variety of other immune activation markers through flow analysis. Authors suggest the increase in immune visibility is mediated through the DNA-damage associated cGAS-STING pathway. When STING KO cells are treated with 6TdG it is suggested that MHC-I and the NK activation marker, NKG2DL, is decreased. However, the method of measurement is unclear and therefore impossible to independently interpret. The STING KO cell line was used in vivo and authors observed a loss of efficacy in 6TdG treatment. The authors investigate the therapeutic efficacy of 6TdG in combination with IR, to investigate a more clinically relevant treatment strategy. Authors observed a strong decrease in tumor burden when 6TdG is combined with IR compared to single agent treatment. This manuscript describes an intriguing story and the strong reduction in tumor size with combination of 6TdG with IR has the potential to add a novel therapeutic strategy to SCLC, which desperately needs clinical advancements.

Major concerns about author's conclusions:

“This treatment also decreased both primary and distal metastasis from the subcutaneous tumor by depleting low antigen presenting CICs.”

- Authors show data that suggest 6TdG treatment has increased efficacy when cells or engrafted tumors are enriched with CICs. However, there is no data presented that in vivo treatment of 6TdG depletes number of CICs in bulk tumor experiments. Therefore, this conclusion is premature. SCLC tumors have a high number of cells with CIC qualities, I would like to see an experiment where a tumor initiated by non-sorted cells was dissociated after treatment and analyzed by flow for CIC marker expression in order to support this conclusion.

“Telomere targeting in vivo led to activation of the CD8 T cells and NK cells against tumor cells dependent on tumor STING signaling.”

- Authors show data that there is an increased expression of CD3, a T cell activation marker, after 6TdG treatment but do not show any T cell activation experiments. Without these experiments CD3 expression is only a correlative observation for CD8 T cell activation. In addition, some of the initial in vivo experiments showing 6TdG efficacy was done in Nu deficient mice (Jax 007850) which have a deficiency in T-cell function. How do you explain the efficacy observed in this mouse model with the statement that CD8 and NK activation is necessary for 6TdG anti-tumor function?

- Authors suggest that 6TdG activity is dependent on STING function within the tumor cells and engrafted 984-STING KO tumors which did not appear to respond to 6TdG treatment. However, authors do not provide sufficient support for their hypothesis. Authors suggest STING activity is initiated through 6TdG associated DNA-damage that triggers the cGAS-STING pathway. Authors do not look at any activation markers in the cGAS-STING pathway. STING is a potent activator of IFN-I, authors show IFN-alpha and IFN-beta measurements, but do not measure any interferon-stimulated genes (ISGs), which is standard practice for both STING and IFN studies. In addition to missing important support for this claim, the data they did show looking at IFN-alpha and IFN-beta levels was lacking any experimental information. It is completely unacceptable to not even mention what biological material is being measured.

Major concerns or questions related to the data shown:

- Poorly organized and written in a non-cohesive manner. Strongly suggest increased proof

reading, there was clearly two voices in the authorship and minor but frequent errors.

- Significant lack of information in the Materials and Methods section. Authors only included mouse strain information on Nu mice, no information on the wild type strain. Data shown in both the main text and supplemental information have a severe deficiency in experimental information, to the extent that it is difficult to impossible to interpret the data. Some examples of experiments with insufficient or no information include the cell viability assay used, how the colony formation assay was quantified, how the statistical analysis was performed in the tumor volume measurement (curve vs endpoint comparison), anything on the qPCR-based assay for telomerase activity, in vitro sphere formation assay, or the CD8 or NK depletion method, and these are just examples from the first half of the manuscript.

- In addition to the lack of experimental details, many figures are missing units or other necessary information to properly interpret the information presented.

- Authors claim that 6TdG was more efficient than standard chemotherapy against 5 cell lines, however they only show data on two cell lines. In addition, the human SCLC data they present shows a minimal difference in the cell survival curves. Authors claim a significant difference in the IC50 values but this is not a correct evaluation, if anything the authors could compare the survival curves.

- Authors investigate an induction of senescence but do not comment on the commonalities between triggering senescence and apoptosis.

- Only one cell line is shown for the senescence experiments, and appears to be 987A. Based on the information provided by the authors, it appears that 987 and 987A are derived from the same tumor but one is grown in suspension and 987A is grown adherently. Do the authors see the same senescence observations in both 987 and 987A or is this observation due to culturing conditions? Especially since there is a large difference in IC50s reported in 1B. This question also applies the panel 2E when authors compare 987 and 987A CIC vs non-CIC populations.

- In panel 1H tumor burden is compared between standard of care chemotherapy, cisplatin-etoposide, and 6TdG treatment. How the statistical analysis performed and how many animals was were used? It appears to be a modest difference for a ** significance.

- What cell line was used in figure 2B?

- What is exactly being measured in this qPCR-based assay for telomerase activity in Figure 2D?

- Figure 2E shows colony growth in two cell lines. Figure legend states that colony surface area was measured but none of the figures in this panel appear to reflect that. 987 is measured by tumor sphere number and the H69 graph is not interpretable, a relative cell number of 4?
- Panel 2F: What cell line or populations were used? Is the bar graph quantifying information from the heat map or a separate experiment looking at MHC-I levels in previously sorted cells?
- Gross histology H&E figures of mouse livers, the stain is incredibly dark and in some figures completely masks the tumor. Some of the figures (ex 3H) appear to have the contrast increased and make it even more difficult to see the tumors.
- In many of the in vivo experiments, the time line describes that treatment starts after tumors are established but how do the authors determine if the tumors are established? Is there an imaging mechanism to determine tumor presence or is it a set time post IV injection of SCLC cells?
- The in vivo data shown should have the N listed but most figures/ figure legends do not say.
- The microscopy images should have scales included.
- Many graphs are missing the units of measurement.
- The mouse strain used for experiments shown in Figure 4 is unclear. In addition all the experiments appear to be evaluated in different ways, tumor volume, lesion number, and liver/body weight. Would like to see a consistent measurement in order to evaluate each experiment.
- Figure 5A is completely unclear what cell populations the authors are measuring, needs more sufficient labeling.
- Some of the figures (ex. 5D) show multiple data points but we are not told what these points represent. Are they technical replicates? More information is needed.
- Figure 5F shows bar graphs of IFN alpha and beta levels but it is not disclosed material was measured. In addition, increased IFN production should be verified by including gene expression data of ISGs.
- IFN can be induced in ways other than through STING, and STING can be activated in other ways than cGAS. I would like to see verification that this is actually the signaling cascade that is being activated.

- I would like to see a comparison of STING KO vs vector control with or without 6TdG treatment (figure 5H)
- Figure 6 is a lot of information showing an induction of activating markers for T cells. Are the T cells actually being activated?
- Figure 7A claims synergy between 6TdG and IR, was synergy actually calculated? If so, what was the combination index?
- Figure 7C depicts a KM survival curve. How many animals were in each group? What was the endpoint criteria? It seems suspicious that all the animals in the control group reached endpoint on the same day.
- Sup Figure 1C: Shows relaxed gating “to be able to collect enough numbers” what was the stringent gating? Was there a target number for each group, and if so, what is that number?
- Authors write “C8T cell recruitment was also increased in the combination group (Supplementary Fig 4)” Technically the statement is true but it is misleading. There does not appear to be a difference between the combination group and 6TdG single agent group. The statistics between the control group and the single agent group was not performed. I would interpret this data as the 6TdG treatment has an effect on T-cell migration, not the combination treatment.

Reviewer #4 (Remarks to the Author): with expertise in lung cancer immunology, cancer initiating cells

Manuscript by Buse Eglenen-Polat “Depleting cancer initiating cells induces immune visibility and anti-tumor immunity in small cell lung cancer”

The authors test the potential of the nucleoside guanine analogue 6-Thio-2'-deoxyguanosine (6TdG), which incorporates into de novo-synthesized telomere causing dysfunction and results in inhibition of SCC tumor growth and metastasis. The purported mechanism suggested was that 6TdG targets cancer stem cells and activated STING to unleash adaptive immunity. The study is significant as 6TdG has potential to emerge as a new treatment approach for SCC. However, given the failure of several telomeres inhibitors in the past, key issues need to be addressed for publication of this work in Nature communications.

Major Comments:

1. A major focus is on the L1CAM+ “cancer stem cells” in SCC which were shown to have increased telomerase activity, low MHCI expression and increased metastatic potential (i.v injection) in vivo compared to L1CAM- cells. However, these cells need not been properly characterized. For in vivo metastasis studies, can these cells be tracked in vivo from the primary tumor site to the metastatic organ using a lineage tracing approach. This would allow determination on which steps of the metastatic cascade is impacted the most by the drug.
2. In the subcutaneous model (Fig. 3), its suggested that 6TdG reduced distant metastasis. Is this because primary tumors were significantly impaired or 6TdG selectively targeted the telomerase high cancer stem/metastatic cells.
3. Figure 2 and related text mentions that 6TdG led to lower L1CAM expression and decreases cancer initiating cells. What happens to these CICs- are they differentiated after treatment or undergo apoptosis? It is important to understand what happens to CICs upon treatment with 6TdG.
4. While telomerase+ cells are sensitive to 6TdG, it is important to determine the impact of 6TdG on non-cancer initiating cells? Also telomerase activity is a feature of normal stem cells, for instance HSCs in bone marrow and other normal stem cells. What is the impact of 6TdG in the models that the authors used on these normal populations?
5. It is expected that 6TdG would compete with guanine nucleotide and incorporate possibly at lower levels during normal DNA replication. Was this evaluated? Do the cells repair this.
6. The investigation on the immune component is underdeveloped (See below)
7. Fig 4, why does CD8 T cell depletion reduce the therapeutic efficacy of 6TdG, given that 6TdG is expected to target MHC-I low telomerase+ cancer stem cells which are likely to be targeted by NK cells. Moreover, treatment increases NK cell activating ligands.

8. Authors talk about increased Granzyme B and IFN γ in T-cells, but this difference is significantly reflected only in total CD3 population not in CD8 T-cells specifically (supplementary figure 2 SB). What is the source of Granzyme B and IFN γ what subset of CD3 T cells are producing these cytokines? What is the impact on TNF α and PD1 expression on CD8 T cells?
9. What is the impact of co-culture between CD4 T cells and 6TdG-pretreated SCLC cells?
10. Figure 6B: Are these absolute numbers for different subsets from single cell RNA-seq data or flow cytometry? It may not be ideal to use cell numbers to quantify differences in cell populations from scRNA-seq.
11. Which MDSC population in liver (Figure 6F) is impacted by 6TdG treatment- PMN-MDSC or M-MDSC?
12. Figure 6F shows no difference in NK cell numbers after 6TdG treatment. Does 6TdG treatment only impact cytokine secretion from NK cells (as shown in co-culture experiment in supplementary Fig 2SB)? Did the authors also evaluate IFN γ from NK cells in *in vivo* model?
13. In the *in vivo* model in Figure 6, what is the impact of treatment on T-cell activation (IFN γ , TNF α , granzyme B), proliferation (Ki67 and PD-1 expression)?
14. What is the mechanism by which single agent 6TdG increase STING. Is there involvement of dsDNA, TREX etc.
15. It would be important to know if 6TdG is a better option to existing telomerase inhibitors, particularly in the context of combination with immunotherapy.

Minor comments:

1. Details on co-culture experiments, how they were performed can be expanded in methods section?
2. For the radiation-6TdG combination therapy, please show a schema for tumor injection and treatment timeline in figure 7.
3. Provide details on statistical analysis for every figure section individually in the figure legend.
4. Both B6129SF1/J and nu/nu mice are used with 984 model at different points in the study. Please label in the manuscript as well as figure legend what strain is used in a certain experiment.
5. Minor grammatical errors:
 - a. Abstract “SCLC models that were dependent on immune cells”
 - b. Methods “B6129SF1/J mice or nu/nu mice”

We thank the reviewers for their time and constructive feedback. We made major edits to the manuscript based on their recommendations. These include:

- 1-Major edits to the figures and text, inclusion of new higher resolution microscope images, graphs, and plots
- 2-Major edits to the legends and methods to include relevant information
- 3-In vitro experiments to show increased MHC-1 and type-I interferons and IFN induced genes with additional cell lines and L1CAM/CD133 sorting experiments.
- 4-Co-culture experiments with isolated donor CD4 or CD8 T cells and measurement of secreted cytokines
- 5-Suggested mouse experiments:
 - Vector control and *Sting* ko mouse cells
 - Telomerase inhibitor imetelstat experiment,
 - Combination treatment with PD-L1 blockade,
- 6-Strengthening of telomere connection by including additional experiments including mouse therapy experiment with *Tert* KO knockout SCLCs
- 7-Strengthening data regarding type-I interferon activation, IFN gene expression in SCLCs after treatment with 6TdG

We believe we addressed all the comments raised by the reviewers and new results strengthened our conclusions. We hope that the reviewers find our revised manuscript acceptable for publication.

Point by point responses to Reviewer comments:

Reviewer #1 (Remarks to the Author): with expertise in lung cancer immunology, SCLC, STING
In this manuscript, Eglenen-Polat and Colleagues describe a new potential therapeutic approach for small cell lung cancer (SCLC) by treating with a nucleoside analog, 6-Thio-2'-deoxyguanosine 6TdG. They find that inducing telomere dysfunction can efficiently inhibit primary SCLC tumor growth and metastasis from the primary tumors, preferentially depleting cancer initiating cells with higher telomerase activity. The authors suggest that tumor control is mediated by NK and the T cells and observe a synergistic effect of telomere damage therapy with radiotherapy in SCLC. The authors suggest that this anti-tumor effect is tumor-STING dependent, however, the data does not fully support this conclusion with strong evidence. While the concept has several implications for small cell lung cancer patients and is potentially interesting for the readers of this journal, additional work is needed to substantiate their claims, particularly with respect to the data presented in Fig. 5.

Major points:

1. Figure 5A, C: The authors showed that 6TdG treatment for three days induces surface MHC-I expression by flow cytometry in mouse SCLC lines in a dose dependent manner and in one human cell line (H69). Have the authors examined induction of surface MHC-I expression in additional SCLC lines to recapitulate the same effect?

Response 1- Two other human cell lines H510, and H2081 were also treated with vehicle or 1uM 6TdG. Consistent with 984 and H69 cells, we observed significantly increased MHC-1 expression compared to the vehicle treated cells. These data are included as revised figure 4f and supplementary figure 3c.

2. Figure 5B: Regarding studies of NK cell activating ligands-NKG2DL in mouse SCLC cells - have the authors examined activating and inhibitory ligands on the surface of several SCLC human lines?

Response 2- NKG2DL expression was included in the previous submission. To address reviewer's comment, we additionally stained human SCLC cells for DNAM1 ligands (DNAM1-Fc), HLA-E and HLA-G. There are significant increases in all of the three ligands, although % of HLA-E AND HLA-G positive cells are low in SCLC. The figure representing these data are shown below and included in revised figure 4 and Supplementary Figure 3c. We believe that these changes may have different influence on regulating NK/T cell activity resulting in overall more activation than inhibition and ultimately tumor cell killing based on our in vitro (PBMC culture and mouse immune phenotyping experiments as detailed in response to other comments).

3. Figure 5C: Could the authors explain which NKG2D ligands that they were screening for in H69? MICA/B?

Response 3- NKG2DL expression was determined by using NKG2D-Fc fusion proteins: mouse NKG2D-Fc (R&D systems catalogue # 139NK050) and human NKG2D-Fc (Sinobiological catalogue # 10575-H01S-50) or specifically MICA/B antibody or Raed antibody where indicated. This information is indicated on the figure legends now. NKG2D-Fc fusion protein detects all ligands binding to the NKG2D receptor (MICA/B and ULBPs).

4. Figure 5D: The methods/conditions for these experiments are unclear, and the figure legend does not explain if intracellular flow staining was utilized. PBMCs cells were used and then gated on CD3+ cells. It would be nice to repeat this experiment using isolated CD3+ T cells for co-culture, also measuring secreted cytokines such as IFN γ and Granzyme B ELISA to support their claims.

Response 4- Methods were updated to improve clarity. Intracellular staining protocol detailed in the methods was used to measure IFN γ and granzyme B by flow cytometry. This information is now included in the figure legend.

To determine secreted levels of cytokines we repeated the co-culture experiment with either whole PBMCs or CD3+ PBMCs. Vehicle or 6TdG pre-treated H69 or H510 cells were co-cultured with PBMCs. Data below shows flow cytometry results when gated on CD8 T cells and ELISA is from the co-culture media. Granzyme B and IL2 expression is significantly increased in CD8+ PBMCs when co-cultured

with 6TdG pre-treated H510 cells. Granzyme b is significantly increased when PBMCs were co-cultured with 6TdG pre-treated H69 cells. As for secreted cytokines, Granzyme B and IFN γ are increased in H510 co-cultures and granzyme B secretion is increased in H69 co-culture. Indicating activation of T cells when co-cultured with pre-6TdG treated SCLCs. These data are included as revised figure 4g and supplementary figure 3d.

As per the reviewer, we also isolated CD3+ T cells from whole human PBMC's using isolation beads. Vehicle or 6TdG treated H69 or H510 cells were co-cultured with CD3+ T cells. Consistent with whole PBMC results, IFN γ , and IL2 were increased in CD8+ T cells in the H510 co-cultures. In H69 co-culture we also observed significantly increased expression of Granzyme B. This data is shown as revised supplementary figure 3e.

5. Figure 5F: IFN γ induction by 6T-dG seems to fall into low levels of detectability. Also, the authors do not mention which mouse cell line was tested. Could the authors screen addition human cell lines to recapitulate this effect in human models?

Response 5- The 984 mouse SCLC cell line was used in the experiment, this was added on the figure and legend now. This experiment is repeated and new data consistent with previous data. Updated data is provided in revised figure 5 and below. As per the reviewer, we screened additional human SCLC cell lines: H69 and H2081 to evaluate the effect on type-I interferon and interferon inducible genes. Cells were treated with vehicle or 6TdG and then RNA profiled by qPCR for the markers shown below. Consistent with mouse cells, expression of both type-I interferons IFNa and IFNb and IFN inducible genes were increased in 6TdG treated cells. These data are included in revised Figure 5b.

6. Figure 5H: To determine the contribution of STING signaling in mediating 6TdG effects, the authors showed a loss of function experiment in one human and mouse cell line. The authors claimed in their abstract that mechanistically “Mechanistically, 6TdG depleted induced type-I interferon signaling by activating tumor STING”. However, they have not directly examined activation of the cGAS-STING pathway and TBK1-IRF3 signaling. Firstly, the authors should validate their claims using additional lines, and mechanistically prove STING activation by immunoblotting for canonical STING pathway proteins (eg pTBK1, pIRF3 and/or pSTAT1). Moreover, STING activation is known to cause chemokine production (such as CXCL10, CCL5) which will enhance immune cell infiltration, however none of these chemokines were screened.

Response 6-

Thank you for the recommendation. We performed western blot for control and 6-TdG treated 984 and an additional human SCLC cell line (H841) and observed increased pTBK1, pSTING and pIRF3 (more obvious in human cells) molecules in the STING pathway as the reviewer suggested. Human cell line data was added as supplementary Figure 3i.

Per the reviewer qPCR was performed to confirm expression of cytokines downstream of STING signaling in 984, H69 and H2081 cell lines treated with vehicle or 6TdG. We observed significantly higher expression of these markers shown below after 72 hours of 6TdG treatment compared to DMSO control. This data was added as figure 5b and shown in response to above comment #5.

Additionally, we also profiled expression of these markers in 984 control and 984 cGAS knockout cells and found significantly higher expression of type-1 interferons and IFN inducible genes. The results indicate activation of and type-I interferon signaling and requirement of cGAS for this phenotype. The figure representing this data is included in revised figure 5d.

Minor points:

7. Figure 3H, statistical analysis is missing in the first panel.

Response 7- Statistical analysis for the first panel of Figure 3h (figure 5e in the revised manuscript), now added to the panel and to the figure description.

8. In the results section, a reference about the first line of SCLC therapy is missing.

Response 8- A reference was added to the corresponding section.

9. Reference to Fig. 5G missing in the manuscript.

Response 9- The following sentence has been added to the text discussing the findings in the graph:

While 6TdG treatment increased MHC1 and NKG2DL expression in wild-type H69 cells, this effect is not observed on STING KO H69 cells treated with 6TdG. This data is now shown in figure 5e in the revised manuscript.

10. The authors should consider moving the introduction on cGAS-STING pathway from results into the introduction section.

Response 10- The paragraph regarding the STING pathway in the result section was moved to introduction section.

Reviewer #2 (Remarks to the Author): with expertise in cancer, telomeres/telomerases

In this study, Eglenen-Polat et al. investigate the chemotherapeutic mechanism of 6-thio-2'-deoxyguanosine (6TdG) as an anti-telomerase activity strategy in small cell lung cancer (SCLC) models, both in vitro and in vivo. They demonstrate that 6TdG treatment exhibits significant anti-tumor activity by inducing DNA damage at telomeres, also referred to as telomere dysfunction-induced foci (TIFs), in cancer initiating cells (CICs) that predominantly depend on telomerase activity for proliferation. The authors investigate the anti-tumor activity and reveal that TIFs subsequently activate the STING pathway, generating a favorable immune response within the tumor microenvironment and leading to the recruitment of T cells and natural killer (NK) cells. Additionally, they extend their findings by illustrating that 6TdG treatment, when combined with ionizing radiation (IR), could serve as an effective alternative for SCLC treatment by exhibiting a synergistic effect in vivo. These results hold relevance for the field and offer a new direction for targeting in specific cancer subtypes, yet I am not convinced that this is through a telomere deprotection pathway. The only panel that speaks to this is 1E. The telomere connection has to be strengthened or removed. The presentation and quality of the data is not up to the standard for Nature Communications.

Major concerns:

1. These results hold relevance for the field and offer a new direction for targeting in specific cancer subtypes, yet I am not convinced that this is through a telomere deprotection pathway. The only panel that speaks to this is 1E. The telomere connection has to be strengthened or removed.

Response 1- To further strengthen this connection as per the reviewer, we performed additional *in vitro* and *in vivo* experiments. We performed an additional TIF assay in a human SCLC cell line. We saw significantly increased number of TIFs (31 fold) with 6TdG as compared to DMSO treated cells. We also observed increase in general DNA damage (19 fold). However, as telomeres are only a small fraction of the genome, if proportional damage to the rest of the genome was expected, increase in total DNA damage would be 6000x more. Through metaphase spreads we observed increased number of dicentric chromosomes fused from telomeres, and telomere fragments resulting from dysfunctional telomeres. These data are included in revised figure 1f-h.

To prove the requirement of telomerase in the therapeutic efficacy with 6TdG, we performed an *in vivo* experiment in which *Tert*^{-/-} mouse SCLC (984) tumors were treated with 6-TdG. In this experiment,

therapeutic efficacy was lost as we did not observe a difference in the tumor growth between the control and treated groups. This data has been added as Figure 1j in the revised manuscript.

2. The presentation and quality of the data is not up to the standard for Nature Communications. In many cases the quality and resolution of the figures presented is poor and does not allow interpretation of the data. This is particularly relevant for IHCs images and immunofluorescence. H&E is overstained, pictures are out of focus, etc. Size bars are generally missing.

Response 2- We appreciate the reviewer comment. In the revised manuscript we have now updated and improved figures, reorganized them and are displaying higher resolution images. We also added scale bars to all the relevant images.

3. Figure 1G is not convincing that this reflected telomere colocalization and qualifies as a TIF.

Response 3- We provide below a new representative TIF image. This is included in revised figure 1f.

3. The authors should specify how statistics have been used in each figure panel. In particular the statistics used behind the p values indicated in the figures is not clear. In addition the authors should indicate how many times each experiment has been performed. The majority of the graphs are difficult to read as in most cases the authors do not specify what the dots in the graph actually represent. We suggest the authors clearly state what dots represent in each graph.

Response 3- We thank the reviewer for the comment. We reviewed the figure legends and added the information regarding the number of repeats performed and statistical analysis method used when relevant. Data represented in graphs is now clearly stated for each figure legend.

4. Figure 1E-F, please indicate how many biological replicates have been used for this panel. In addition, please specify in more detail the statistics used for these figures.

Response 4- In figure 1E in the revised manuscript we show flow cytometry analysis of the percentage of γ H2AX positive 984 cells increasing with the dose of 6TdG. This experiment was done using flow cytometry in triplicates. Control versus treatment samples were analyzed using student's T-test. These statements were added to the legend of figure 1. Each data point from replicates in experiments are displayed in the bar graphs for all the manuscript figures.

5. Figure 1G, the quality of the images presented is low and prevents a correct interpretation of the results. In addition it is not indicated how many times this staining has been performed.

Response 5- We performed a new TIF assay on human SCLC cell line H841. In the revised manuscript, we have changed the images with higher resolution ones shown in Figure 1f. Experiment was repeated twice.

6. Figure 2B, please indicate what the dots in the bar plots represent. Are these 3 independent experiments?

Response 6- The dots in this plot represent the percentage of Ki67+ positive 984 cells measured by flow cytometry sorted for L1CAM negative and positive populations. Data represents triplicates and experiment was repeated twice. This new graph is included in the revised manuscript as figure 2b.

7. Figure 4A-C, the effective depletion of CD8+ and NK cells is not shown. Please provide data showing that this depletion treatment was effective.

Response 7- Efficiency of CD8/NK cell depletion by antibodies in the blood of mice are shown below. Graphs were included in revised supplementary figure 3b.

8. Figure 5B, please indicate clearly what the dots are representing.

Response 8- Each data point represents replicates. This data is now included in revised figure 4e.

9. Page 17: "Therapeutic efficacy of 6TdG was lost: there was no significant difference between metastatic tumor burden of 6TdG treated or control STING-KO tumors, suggesting that in this SCLC model system the therapeutic effects of 6TdG is tumor STING-dependent, an important difference from other immunogenic models and highlights tumor STING as a vulnerability of SCLC". This paragraph misses a reference to a figure. Is this referred to Fig. 5H?

Response 9- This figure has now been moved to figure 5g of the revised manuscript. A reference to this figure has been added.

10. Fig. 6E, the FACS plot graph presented is of poor quality and difficult to interpret. Please provide a higher quality plot for this data.

Response 10- This plot was updated with a higher resolution plot as shown below.

11. Fig. 6G, the FACS plot graphs presented are barely interpretable. Please reorganize these panels and provide a higher quality presentation of this data.

Response 11- These plots have been replaced with resolution plots, this data is included as revised figure 6g.

12. Fig. 3K, possibly erroneous figure call. It looks like this is actually referred to Fig. 6H.

Response 12-This reference has been corrected to reflect the correct figure.

13. Fig. 7A, this data are better represented as dotted lines as in previous panels, please modify it to dotted lines representation.

Response 13- Data was changed to dotted line presentation as below and included in the revised manuscript as figure 7a.

Reviewer #3 (Remarks to the Author): with expertise in SCLC, tumor initiating cells, STING This study by Eglenen-Polat and Zhu et al. depicts the therapeutic potential of telomere-targeting agent, 6TdG in small cell lung cancer (SCLC). The authors show that 6TdG treatment causes cell death in vitro and increased DNA damage both in vitro and in vivo. Authors also show that there is a decrease in tumor burden in 6TdG compared to Cisplatin-etoposide treated mice, however the difference is modest and the method of statistical analysis performed is not disclosed. When SCLC populations are enriched for cancer initiation cells (CICs), 6TdG treatment shows increased cell death in vitro and decreased

tumor burden in vivo. Authors conclude that 6TdG treatment target CICs in vivo. Authors investigate a previously described observation that 6TdG treatment induces an immune response in vivo, by depleting CD8 or NK cells and show a loss of 6TdG efficacy in regards of tumor burden. To further explore this observation, authors show that there is an increase in MHC-I expression after 6TdG treatment in vitro and an increase of a variety of other immune activation markers through flow analysis. Authors suggest the increase in immune visibility is mediated through the DNA-damage associated cGAS-STING pathway. When STING KO cells are treated with 6TdG it is suggested that MHC-I and the NK activation marker, NKG2DL, is decreased. However, the method of measurement is unclear and therefore impossible to independently interpret. The STING KO cell line was used in vivo and authors observed a loss of efficacy in 6TdG treatment. The authors investigate the therapeutic efficacy of 6TdG in combination with IR, to investigate a more clinically relevant treatment strategy. Authors observed a strong decrease in tumor burden when 6TdG is combined with IR compared to single agent treatment. This manuscript describes an intriguing story and the strong reduction in tumor size with combination of 6TdG with IR has the potential to add a novel therapeutic strategy to SCLC, which desperately needs clinical advancements.

Major concerns about author's conclusions:

1. "This treatment also decreased both primary and distal metastasis from the subcutaneous tumor by depleting low antigen presenting CICs."

- Authors show data that suggest 6TdG treatment has increased efficacy when cells or engrafted tumors are enriched with CICs. However, there is no data presented that in vivo treatment of 6TdG depletes number of CICs in bulk tumor experiments. Therefore, this conclusion is premature. SCLC tumors have a high number of cells with CIC qualities, I would like to see an experiment where a tumor initiated by non-sorted cells was dissociated after treatment and analyzed by flow for CIC marker expression in order to support this conclusion.

Response 1- As the reviewer suggested, short-term treatment of subcutaneous tumors with 6TdG also caused a decrease in L1CAM+ Epcam+ cells. This data was added to supplementary figure 5b in the revised manuscript and details were added to the legends.

2. "Telomere targeting in vivo led to activation of the CD8 T cells and NK cells against tumor cells dependent on tumor STING signaling."

- Authors show data that there is an increased expression of CD3, a T cell activation marker, after 6TdG treatment but do not show any T cell activation experiments. Without these experiments CD3 expression is only a correlative observation for CD8 T cell activation. In addition, some of the initial in vivo experiments showing 6TdG efficacy was done in Nu deficient mice (Jax 007850) which have a

deficiency in T-cell function. How do you explain the efficacy observed in this mouse model with the statement that CD8 and NK activation is necessary for 6TdG anti-tumor function?

Response 2-We appreciate the reviewer comment. To further demonstrate the effect of 6TdG on T cells NK cells we performed the following *in vitro* and *in vivo* experiments detailed below.

Vehicle or 6TdG pre-treated H69 or H510 cells were co-cultured with PBMCs. Data below shows flow cytometry results when gated on CD8 T cells and ELISA is from the co-culture media. Granzyme B and IL2 expression is significantly increased in CD8+ PBMCs when co-cultured with 6TdG pre-treated H510 cells. Granzyme b is significantly increased when PBMCs were co-cultured with 6TdG pre-treated H69 cells. As for secreted cytokines, Granzyme B and IFN γ are increased in H510 co-cultures and granzyme B secretion is increased in H69 co-culture. Indicating activation of T cells when co-cultured with pre-6TdG treated SCLCs. These data are included as revised figure 4g and supplementary figure 4d.

We also isolated CD3+ T cells from whole human PBMCs. Vehicle or 6TdG treated H69 or H510 cells were co-cultured with CD3+ T cells. CD3+ T cells were harvested for flow cytometry. Consistent with whole PBMC results, granzyme B, IFN γ , and IL2 were increased in H510 co-cultures. In H69 co-culture we observed significantly increased expression of Granzyme B. This data is shown as revised supplementary figure 4e.

6TdG treatment caused activation of CD8 and NK cells *in vivo* too. We observed significantly increased granzyme B, IFN γ and Ki67 expression among CD8+ T cells in 6-TdG-treated tumors compared

to vehicle-treated controls. CD107a expression was higher among CD8+ T cells in the treatment group. IFN gamma expression was also increased among NK cells in treated subcutaneous tumors. CD8T cell and NK cell data have been added as supplementary figure 5b in the revised manuscript. We revised the result description to say “CD8 and NK cells contribute to the response to 6TdG treatment in vivo.”

3- Authors suggest that 6TdG activity is dependent on STING function within the tumor cells and engrafted 984-STING KO tumors which did not appear to respond to 6TdG treatment. However, authors do not provide sufficient support for their hypothesis. Authors suggest STING activity is initiated through 6TdG associated DNA-damage that triggers the cGAS-STING pathway. Authors do not look at any activation markers in the cGAS-STING pathway. STING is a potent activator of IFN-I, authors show IFN-alpha and IFN-beta measurements, but do not measure any interferon-stimulated genes (ISGs), which is standard practice for both STING and IFN studies. In addition to missing important support for this claim, the data they did show looking at IFN-alpha and IFN-beta levels was lacking any experimental information. It is completely unacceptable to not even mention what biological material is being measured.

Response 3- We apologize for the oversight. We now provided details of the experiments in the legends and on the figures. We are including additional supportive data as below. 6TdG significantly increased micronuclei formation most of which were uncoated, potentially inducing cGAS/STING pathway due to cytoplasmic DNA.

We performed western blot for control and 6-TdG treated mouse and human SCLC cell lines and observed activation of the STING pathway as the reviewer suggested. H841 data was added as supplementary Figure 3i.

As per the reviewer qPCR was performed to determine expression of immunomodulatory genes including IFN α , IFN β , CXCL10 in human SCLC cell lines H69 and H2081 treated with vehicle or 6TdG. We observed significantly higher expression of these markers shown below after 72 hours of 6TdG treatment compared to DMSO control. This data was added as figure 5b in the revised manuscript.

Additionally, we also profiled expression of these markers in *Wt/Cgas* KO 984 cells. All except one of the genes were significantly increased in 984 cells, and this increase was dependent on cGAS. The figure representing this data was added to figure 5d in the revised manuscript.

Major concerns or questions related to the data shown:

- 4 - Poorly organized and written in a non-cohesive manner. Strongly suggest increased proof reading, there was clearly two voices in the authorship and minor but frequent errors.
- 5 - Significant lack of information in the Materials and Methods section. Authors only included mouse text and supplemental information have a severe deficiency in experimental information, to the extent that it is difficult to impossible to interpret the data. Some examples of experiments with insufficient or no information include the cell viability assay used, how the colony formation assay was quantified, how the statistical analysis was performed in the tumor volume measurement (curve vs endpoint comparison), anything on the qPCR-based assay for telomerase activity, in vitro sphere formation

assay, or the CD8 or NK depletion method, and these are just examples from the first half of the manuscript.

Response 4 and 5- We thank the reviewer for the comments. We reorganized the manuscript and corrected deficiencies in writing. We have added detailed protocols for IC50 measurements, colony formation assay, *in vitro* sphere formation assay and qPCR-based assay for telomerase activity were added to the methods section. These sections read as:

Cell viability measurements

An equal number of cells were counted and plated in each well of opaque-walled 96-well plates. Cells were treated with dilutions of 6TdG (10uM, 3uM, 1uM, 0.3uM, 0.1uM, 0.03uM, 0.01uM, 0.003uM, 0.001uM and control) diluted in cell media for 5 days. For *in vitro* experiments a 1mM stock of imetelstat was prepared by diluting the drug in PBS. Cisplatin stock solution was diluted in PBS for *in vitro* assays. This stock was further diluted in media for experiments. Three replicates were performed for each cell line unless otherwise indicated. A luminescent cell viability assay (CellTiter-Glo® Promega # G7570) was used to determine cell viability in each well. The manufacturer's protocol was used when incubating the cells, and plates were protected from light during incubation process. The luminescence was measured using Tecan microplate reader. Survival curves were created and IC50 for each cell line was calculated using Graph Prism.

3D colony formation assay

984 and H69 cells grow in suspension, 984 grows as tight spheres and H69 grows as disorganized clusters. 5×10^3 sorted cells were plated per well, in a 96 well plate and cell growth was observed for 5 days. Wells were photographed daily. Colony surface areas for each well were calculated using Image J version 8.

2D colony formation assay

Adherently growing mouse and human cell lines were used for these experiments. An equal number of cells were sparsely seeded in a 6 well plate. Cells were treated in one third dilutions of 6TDG (10uM, 3uM, 1uM, 0.3uM, 0.1uM and control) in cell media for 7 days. At the end of treatment period, media was aspirated, cells were fixed with 10% formalin, and stained with crystal violet dye (1% crystal violet in 10% ethanol in PBS). The plates were washed with PBS, dried and scanned using BioRAD ChemiDoc. The number of colonies in each well was counted using Image J. Three repeats were performed for each cell line.

qPCR-Based Telomerase Activity Assay

Telomerase Activity Quantification qPCR Assay Kit (TAQ)(ScienCell #8928) was used to quantitatively compare telomerase activity between CIC and non-CIC tumor cell populations. After L1CAM negative and L1CAM positive tumor or CD133 high and CD133 low populations are separated via FACS, the cells were lysed and prepared according to manufacturer's protocol. The cell lysis buffer is a mild lysis buffer that enables the release of telomerase in the native state. The telomere primer set (TPS) recognizes and amplifies newly synthesized telomere sequences in the assay. Quantification cycle value difference between the two samples (ΔCq , Telomere primer set (TPS)) according to manufacturer's instructions for each cell population and was compared to pooled/unsorted tumor cell population. ΔCq (TPS) was calculated as ΔCq (TPS) = Cq (TPS, sample 2) - Cq (TPS, sample 1). Three replicates were performed for each cell line.

Cell depleting antibodies: NK cells were depleted using anti-NK1.1 (BioXcell, anti-mouse NK1.1 clone PK136) or anti-CD8 antibody was depleted using anti-mouse CD8a (BioXcell, anti-mouse CD8a, clone 2.43). Antibodies were administered intraperitoneally to the mice at 200ug/dose at the indicated

schedules.

6- In addition to the lack of experimental details, many figures are missing units or other necessary information to properly interpret the information presented.

Response 6- We reviewed and updated all experimental details. Figures now include necessary information to easily interpret data.

7- Authors claim that 6TdG was more efficient than standard chemotherapy against 5 cell lines, however they only show data on two cell lines. In addition, the human SCLC data they present shows a minimal difference in the cell survival curves. Authors claim a significant difference in the IC50 values but this is not a correct evaluation, if anything the authors could compare the survival curves.

Response 7- Per the reviewer we tested efficacy of 6TdG in comparison to cisplatin in two other human SCLC lines H1048 and H510. We agree that the difference is not an order of magnitude in all the lines. We included data with the two new lines and revised the results as "We evaluated the efficacy of 6TdG in comparison to cisplatin. SCLC cells tested - H1048, H69, and H510 (Figure 1a) and mouse SCLC cell line (984) were slightly but significantly more sensitive to 6TdG than cisplatin". In the legend of Figure 1a and 1b, it has been reiterated that curves are being compared.

8- Authors investigate an induction of senescence but do not comment on the commonalities between triggering senescence and apoptosis.

Response 8- 6TdG induced both senescence and apoptosis in SCLC cells, this is indicated in the revised manuscript. We performed an additional experiment where we treated mouse SCLC cell line with vehicle or different doses of 6TdG. There are significantly higher percentage of Annexin positive cells within L1CAM+ cells with 6-TdG treatment as compared to L1CAM-. These data indicate increased apoptosis among tumor initiating L1CAM+ cells with 6TdG treatment. This data has been added as supplementary figure 2a in the revised manuscript.

9- Only one cell line is shown for the senescence experiments, and appears to be 987A. Based on the information provided by the authors, it appears that 987 and 987A are derived from the same tumor but one is grown in suspension and 987A is grown adherently. Do the authors see the same senescence observations in both 987 and 987A or is this observation due to culturing conditions? Especially since there is a large difference in IC50s reported in 1B. This question also applies the panel 2E when authors compare 987 and 987A CIC vs non-CIC populations.

Response 9- The protocol for senescence assay requires cells to grow as adherent cells. Therefore at the moment we are not able to perform this assay on suspension cells and evaluate whether sensitivity correlates with differences in senescence in these cells. A detailed method for the senescence assay was added to the methods section in the revised manuscript.

We repeated the senescence assay with additional human SCLC lines and repeated 984A. Number of senescent cells are significantly increased with 6TdG treatment in SCLC cell lines H841, H510 and 984A cells. These data are in shown in figure 1d and supplementary figure 1d in the revised manuscript.

10- In panel 1H tumor burden is compared between standard of care chemotherapy, cisplatin-etoposide, and 6TdG treatment. How the statistical analysis performed and how many animals was were used? It appears to be a modest difference for a ** significance.

Response 10- We thank the reviewer for catching this mistake. For clarity, we compared final measurements between control, 6TdG and cis-eto and indicated this on the graph. In the revised manuscript this figure is now shown as 1i. The number of mice included in each experiment group was added in the figure description. The statistical tests and level of significance was added to figure legends as appropriate. Specifically for figure 1h, 12 vehicle-treated mice, 12 cisplatin-etoposide treated mice and 8 6-TdG treated mice were analyzed. The graph has been updated to reflect differences between control and treated groups.

11- What cell line was used in figure 2B?

Response 11- 984 cells were used. This is now added to the figure legend for Fig 2b.

12- What is exactly being measured in this qPCR-based assay for telomerase activity in Figure 2D?

Response 12:

Relative telomerase activity was calculated for each cell population by a realtime quantitative PCR based assay relying on telomerase in the tumor cell lysate and telomere primer set. The cell lysis buffer in the kit is a mild lysis buffer that enables the release of telomerase in the native state. The telomere primer set (TPS) recognizes and amplifies newly synthesized telomere sequences in the assay. The following first section was added to the results describing this figure and second section to the method section.

To results section-“Relative telomerase activity of different cell populations were determined by a realtime quantitative PCR based assay relying on telomerase in the tumor cell lysate and telomere primer set in the kit amplifying newly synthesized telomere sequences.”

To methods- “Telomerase Activity Quantification qPCR Assay Kit (TAQ)(ScienCell #8928) was used to quantitatively compare telomerase activity between CIC and non-CIC tumor cell populations. After L1CAM negative and L1CAM positive tumor or CD133 high and CD133 low populations are separated via FACS, the cells were lysed and prepared according to manufacturer’s protocol. The cell lysis buffer is a mild lysis buffer that enables the release of telomerase in the native state. The telomere primer set (TPS) recognizes and amplifies newly synthesized telomere sequences in the assay. Quantification cycle value difference between the two samples (ΔCq , Telomere primer set (TPS)) according to manufacturer’s instructions for each cell population and was compared to pooled/unsorted tumor cell population. ΔCq (TPS) was calculated as ΔCq (TPS) = Cq (TPS, sample 2) - Cq (TPS, sample 1). Three replicates were performed for each cell line.”

The descriptions were added to the figure legends for Figure 2e which now reads as: “Relative telomerase activity (ΔCq) as determined by qPCR assay based on telomerase in cell lysate and telomere primer set. (ΔCq) was determined in sorted L1CAM^{neg} and L1CAM^{pos} 984 and H2081 cells, and CD133^{low} and CD133^{high} H69 cells, compared to unsorted pool.” Figure 2d is now figure 2e in the revised manuscript.

13- Figure 2E shows colony growth in two cell lines. Figure legend states that colony surface area was measured but none of the figures in this panel appear to reflect that. 987 is measured by tumor sphere number and the H69 graph is not interpretable, a relative cell number of 4?

Response 13- We repeated these experiments to achieve better images and wrote a better description. Images have all been updated in the revised manuscript. We quantified the colony coverage area per field for each experiment. This was a better method of quantification as these cells grow in suspension

as the reviewer points each of those clusters appearing as one cell are actually >50 cells especially for H69. In the revised manuscript we are additionally providing data with L1CAM +/- sorted H69. L1CAM sorting of H69 was also consistent with CD133 and positive cells grew larger colonies than L1CAM- cells. The figure legend was corrected to reflect this and now reads as: "Colony area (% coverage of field) for L1CAM^{negative} and L1CAM^{positive} 984 and H69 cells, and CD133^{low} and CD133^{high} H69 cells quantified on the graphs. These new images and quantification are now shown in figure 2f in the revised manuscript. The gating strategy is shown in supplementary figure 2b.

Revised in vitro growth data

14- Panel 2F: What cell line or populations were used? Is the bar graph quantifying information from the heat map or a separate experiment looking at MHC-I levels in previously sorted cells?

Response 14-. Unsorted 984 cells were used in the experiments. MHC1 expression in L1CAM+ and L1CAM- 984 cells were measured by flow cytometry. This information was added to the figure legend which now reads as: "Representative flow cytometry plot of MHC-1 expression in unsorted L1CAM^{pos} and L1CAM^{neg} 984 cells (left). MHC1+ (% of live cells) in unsorted L1CAM^{pos} and L1CAM^{neg} 984 cells per flow cytometry (means 29.6% vs 39.7%) (right)." This data is shown in figure 2g in the revised manuscript.

15- Gross histology H&E figures of mouse livers, the stain is incredibly dark and in some figures completely masks the tumor. Some of the figures (ex 3H) appear to have the contrast increased and make it even more difficult to see the tumors.

Response 15- We scanned the liver slides with a different scanner and replaced images in Figures 2h, 3b, 3j, 4b, 4c. Examples of the new images are shown below.

16- In many of the in vivo experiments, the timeline describes that treatment starts after tumors are established but how do the authors determine if the tumors are established? Is there an imaging mechanism to determine tumor presence or is it a set time post IV injection of SCLC cells?

Response 16- 10 day timepoint was determined based on prior experience with this model. We show below liver tissues with tumors from 5 different mice, 10 days after injection of tumor cell implantation. One of the representative images was added as supplementary figure 3a.

10 day after tumor IV implantation liver tissues

17- The in vivo data shown should have the N listed but most figures/ figure legends do not say.

Response 17- The number of mice used for each experiment were added to legends of relevant figures.

18- The microscopy images should have scales included.

Response 18- Scales were included for microscopy images in the revised manuscript.

19- Many graphs are missing the units of measurement.

Response 19- The figures legends were revised and unit measurements were added to legends of relevant figures.

20- The mouse strain used for experiments shown in Figure 4 is unclear. In addition, all the experiments appear to be evaluated in different ways, tumor volume, lesion number, and liver/body weight. Would like to see a consistent measurement in order to evaluate each experiment.

Response 20- B6129SF1/J hybrid mice (Jax #101043, commonly used strain for syngeneic SCLC mouse experiments) . This information was now added to the figure legend. Each metastatic experiment is now shown as liver/body weight ratio for consistency.

21- Figure 5A is completely unclear what cell populations the authors are measuring, needs more sufficient labeling.

Response 21- 984 cells were used for the experiment. This information was included in the figure legend.

22- Some of the figures (ex. 5D) show multiple data points but we are not told what these points represent. Are they technical replicates? More information is needed.

Response 22- The data points represent the number of replicates in this experiment showing that MHC-1 expression increased with increasing 6TdG concentration in 984 cells. Updated figure with cell information is shown in figure 4d of the revised manuscript.

23- Figure 5F shows bar graphs of IFN alpha and beta levels but it is not disclosed material was measured. In addition, increased IFN production should be verified by including gene expression data of ISGs.

Response 23- IFN alpha and IFN beta gene expression were determined by qPCR. Figure legend and figure now includes this information (fold mRNA relative to control).

qPCR was performed to determine expression of type-I interferons and interferon stimulate genes: IFNa, IFNb, CXCL10 in human SCLC cell lines H69 and H2081 treated with vehicle or 6TdG. We observed significantly higher expression of these markers shown below after 72 hours of 6TdG treatment compared to DMSO control. This data was added as figure 5b.

24- IFN can be induced in ways other than through STING, and STING can be activated in other ways than cGAS. I would like to see verification that this is actually the signaling cascade that is being activated.

We performed western blot for vehicle or 6-TdG treated mouse and human SCLC cell lines and observed activation of the STING pathway as the reviewer suggested. Human cell line data was added as supplementary Figure 3i.

We profiled expression of these markers in 984 control and 984 cGAS knockout cells and found significantly higher expression of multiple such genes in 984 cells compared to cGAS knockout 984 cells. The results indicate the role of cGAS/STING pathway activation in inducing gene expression. The figure representing this data were added to figure 5d.

25- I would like to see a comparison of STING KO vs vector control with or without 6TdG treatment (figure 5H)

Response 25- To address this comment, wild-type mice were injected either vector control or *Sting*-KO 984 cells. While 6-TdG significantly decreased liver tumor burden in mice injected with wild-type 984, no significant change was observed in treatment and control groups in STING-KO 984 tumors. These experiments support that the response to 6TdG treatment is dependent on tumor STING. This data was added as figure 5g in the revised manuscript.

26- Figure 6 is a lot of information showing an induction of activating markers for T cells. Are the T cells actually being activated?

Response 26- There is a significant increase in granzyme B, IFN gamma and Ki67 expression among CD8+ T cells in 6-TdG-treated subcutaneous tumors compared to vehicle-treated controls. CD107a expression was also higher among CD8+ T cells in the treatment group. IFN gamma expression was also increased in CD4 T cells. CD8 T cell data is added as supplementary figure 5b in the revised manuscript.

7- Figure 7A claims synergy between 6TdG and IR, was synergy actually calculated? If so, what was the combination index?

Response 27- In original manuscript we wrote "there is no synergy in vitro". To avoid confusion we removed this sentence.

28- Figure 7C depicts a KM survival curve. How many animals were in each group? What was the endpoint criteria? It seems suspicious that all the animals in the control group reached endpoint on the same day.

Response 28- Thank you for the comment. In the previous experiments mice were euthanatized when treated mice already lived twice as long as the control group. This experiment was repeated to generate a more accurate curve per the reviewer. There were 5 animals in each group. We repeated this experiment, mice were treated as in figure 7b and euthanatized when tumors reach the endpoint criteria.

Consistent with previous data, survival curve demonstrates a significant increase in survival with 6TdG+IR treatment compared to control treated animals. The Kaplan Meier curve of the experiment was shown in Figure 7c and information was clarified in the figure legend.

29- Sup Figure 1C: Shows relaxed gating “to be able to collect enough numbers” what was the stringent gating? Was there a target number for each group, and if so, what is that number?

Response 29- Thank you. A minimum of 1×10^5 sorted cells per mice are needed to establish metastatic tumors in mice based on our experience, only $\sim 1 \times 10^3$ cells are needed for in vitro assays. Only around 10-15% of 984 cells are L1CAM negative and $\sim 90\%$ are positive, it was technically challenging to get enough negative cells for an in vivo experiment with positive-negative gating during one sorting session. Therefore for *in vivo* experiments we sorted L1CAM low and high populations. The number of cells injected for the experiment was detailed in the methods. Gating is shown in supplementary figure 3b of the revised manuscript and methods are detailing these cell numbers.

Gating for cell sorting

30- Authors write “C8T cell recruitment was also increased in the combination group (Supplementary Fig 4)” Technically the statement is true but it is misleading. There does not appear to be a difference between the combination group and 6TdG single agent group. The statistics between the control group and the single agent group was not performed. I would interpret this data as the 6TdG treatment has an effect on T-cell migration, not the combination treatment.

Response 30- CD8 T cell staining was replaced with CD3 staining in the revised manuscript due to much higher number of those in the tumors increasing confidence. A similar trend was observed with CD3 staining. We performed the suggested comparisons, as the reviewer suggests all the treatments

increased CD3+ T cell recruitment. The corresponding sentence was corrected as: "C8T cell recruitment was increased with 6-TdG, IR and combination treatment groups compared to vehicle control tumors.

Reviewer #4 (Remarks to the Author): with expertise in lung cancer immunology, cancer initiating cells

Manuscript by Buse Eglenen-Polat "Depleting cancer initiating cells induces immune visibility and anti-tumor immunity in small cell lung cancer"

The authors test the potential of the nucleoside guanine analogue 6-Thio-2'-deoxyguanosine (6TdG), which incorporates into de novo-synthesized telomere causing dysfunction and results in inhibition of SCC tumor growth and metastasis. The purported mechanism suggested was that 6TdG targets cancer stem cells and activated STING to unleash adaptive immunity. The study is significant as 6TdG has potential to emerge as a new treatment approach for SCC. However, given the failure of several telomeres inhibitors in the past, key issues need to be addressed for publication of this work in Nature communications.

Major Comments:

1. A major focus is on the L1CAM+ "cancer stem cells" in SCC which were shown to have increased telomerase activity, low MHC1 expression and increased metastatic potential (i.v injection) in vivo compared to L1CAM- cells. However, these cells need not been properly characterized. For in vivo metastasis studies, can these cells be tracked in vivo from the primary tumor site to the metastatic organ using a lineage tracing approach. This would allow determination on which steps of the metastatic cascade is impacted the most by the drug.

Response 1: Per the reviewer we labelled RPP cells with a lenti-GFP-luc plasmid to be able to visualize by microscopy (or PCR) and track in vivo by BLI imaging. We sorted cells to enrich for GFP+ cells since this particular dual -GFP-Luc plasmid (the only one we could order) does not have an antibiotic selection marker. For the reasons we do not fully understand cells after sorting do not maintain their suspension phenotype and start adhering. Adherent cells as shown below, have low L1CAM expression and do not metastasize. We also note, there may be other differences between these two.

We performed an imperfect experiment with the unsorted labelled cells to address the reviewer's comment. We did relative quantification of tumor cells by PCR for (GFP and control – Rn18s) from the cells isolated from blood of mice at baseline and after 8 days of treatment with 6TdG. There was a trend towards decrease in GFP signal in the blood of treated mice, however this was not consistent in all the mice. We can't definitively conclude which step is inhibited but consistently we observe less metastasis in 6TdG treated mice.

In an vivo experiment with these labelled cells, we implanted them on both sides to increase circulating tumor cell numbers. We were only able to see BLI signal in the liver in one of the mice at the end point of the experiment even though both mice had tumors in the liver (vehicle treated mice). This is likely GFP negative cells being majority of the growing population.

We were able to isolate an L1CAM low population which grows as pure adherent in culture (RPP-A), as compared to the suspension population. With these two models also we saw a positive correlation with higher L1CAM/CD133 expression and metastasis. Metastasis was only observed in L1CAM high RPP-S cells when they were injected subcutaneously. However, as the reviewer suggests below in comment #2, primary tumor size is also smaller with the L1CAM low cells.

2. In the subcutaneous model (Fig. 3), it is suggested that 6TdG reduced distant metastasis. Is this because primary tumors were significantly impaired or 6TdG selectively targeted the telomerase high cancer stem/metastatic cells.

We thank the reviewer for the comment. To address this question, we waited to sacrifice the mice until tumors are the same size. There is a clear trend towards less metastasis in the treated mice ($p=0.051$, n is not high due to technical limitations of growing these cells within revision timeline after luciferase labelling sorting expanding etc.) as compared to the untreated mice even when primary tumor size is same at the time of euthanasia. As the subcutaneous tumors grow (without follow up treatment, last treatment given around ~20 days before euthanasia), there may be higher chance for metastasis. Follow-up treatments can possibly prevent metastasis since treatment was not toxic as shown in response to comment #4.

We believe it is important to show with another murine model that 6TdG is therapeutically effective in vivo both in IV metastatic model and subcutaneous model. Therefore, we kept the data with RPP model with a revised description of the abstract and results.

In the abstract we revised “Low and intermittent doses of 6TdG inhibited tumor growth and metastasis from the primary tumors.” to “Low and intermittent doses of 6TdG inhibited tumor growth and reduced metastatic burden.”

In the results: “Notably, at the time of euthanasia when controls reached endpoint, metastatic burden was significantly lower in the 6TdG treated mice as compared to vehicle treated mice ($p<0.05$) (Fig. 3j) although we do not know the exact step of metastasis inhibited by 6TdG.”

3. Figure 2 and related text mentions that 6TdG led to lower L1CAM expression and decreases cancer initiating cells. What happens to these CICs- are they differentiated after treatment or undergo apoptosis? It is important to understand what happens to CICs upon treatment with 6TdG.

Response 3- We performed an additional experiment with a mouse line to evaluate the level of apoptosis after 6TdG treatment. We observed a significant increase in the percentage of Annexin V positive cells within L1CAM+ cells with 6-TdG treatment compared to L1CAM- cells. These data indicate higher level of apoptosis among L1CAM+ cells with 6TdG treatment consistent with our hypothesis. This data has been added as supplementary figure 2a in the revised manuscript.

4. While telomerase+ cells are sensitive to 6TdG, it is important to determine the impact of 6TdG on non-cancer initiating cells? Also telomerase activity is a feature of normal stem cells, for instance HSCs in bone marrow and other normal stem cells. What is the impact of 6TdG in the models that the authors used on these normal populations?

Response 4- To address this, we first collected blood from control and 6TdG treated mice. We ran circulating blood cells analysis. There are no differences in the parameters we tested such as white blood cell (WBC) count, lymphocyte numbers, and lymphocyte percentages. These data was added to revised supplementary figure 2b. We consulted with a hematopathologist to evaluate potential damage to the stem cells in the bone marrow. As per his suggestion, we cultured bone marrow cells from either control or 6TdG treated mice in a special viscous bone marrow cell differentiation media-Methocult for 7 days. Hematopathologist saw no differences in overall cell morphology as well as growth rates of HSCs from control or treated mice. Colony forming units were quantified in control and 6TdG treated mice. Representative images are shown below. This data was added to revised supplementary figure 6b and 6c.

5. It is expected that 6TdG would compete with guanine nucleotide and incorporate possibly at lower levels during normal DNA replication. Was this evaluated? Do the cells repair this.

Response 5- We thank the reviewer for this comment. A small fraction of the drug is incorporated by the DNA polymerase, but the telomerase is not as efficient as DNA polymerases in recognizing the difference between 6-thio-dG and guanine, resulting in greater damage at telomeres (/base pair). This

is evident with the increase in TIF vs general double stranded breaks (revised figure 1). Telomeres are only 1/6,000th of genomic DNA, therefore observation that there is 31X more TIFs vs 19X general double stranded breaks suggests that telomerase is more effectively incorporates 6TdG. The presence of telomere-telomere fusions and free telomere fragments indicate damage to the telomeres. Response to comment #4 addresses potential long-term effect on healthy hemopoietic cells and data indicates normal cells are able to recover when mice were treated intermittently with 6TdG.

6. The investigation on the immune component is underdeveloped (See below)

We believe we addressed this comment by strengthening this aspect. Specific responses: T cell, NK cell activation, IFN production, tumor cell IFN inducible gene expression etc were included in response to other reviewers and also below.

7. Fig 4, why does CD8 T cell depletion reduce the therapeutic efficacy of 6TdG, given that 6TdG is expected to target MHC-I low telomerase+ cancer stem cells which are likely to be targeted by NK cells. Moreover, treatment increases NK cell activating ligands.

Response 7- Both NK cells and CD8 T cells contribute to the response to 6TdG. We showed that 6TdG treatment increases MHC-1 expression on remaining live tumor cells. Therefore, this interplay results in CD8 T cell activation as shown in co-culture and in vivo experiments. Since 6TdG also increases NK cell activation. Changes in myeloid cells in the microenvironment could also be contributing to further activation of lymphocytes despite the increase in the inhibitory ligands for example.

There is a significant increase in granzyme B, IFN gamma and Ki67 expression among CD8+ T cells in 6-TdG-treated subcutaneous tumors as compared to vehicle-treated controls. CD107a expression was also higher among CD8+ T cells in the treatment group. NK cells were activated with the treatment as IFN gamma expression was also increased among NK cells in treated subcutaneous tumors. These data have been added as supplementary figure 5a and 5b in the revised manuscript.

8. Authors talk about increased Granzyme B and IFN γ in T-cells, but this difference is significantly reflected only in total CD3 population not in CD8 T-cells specifically (supplementary figure 2 SB). What is the source of Granzyme B and IFN γ what subset of CD3 T cells are producing these cytokines? What is the impact on TNF α and PD1 expression on CD8 T cells?

Response 8- In the mouse experiments, cytokine production is coming from CD4, CD8 and NK cells. As per the reviewer CD8+ T and CD4+ T cells were gated out of the whole CD45+ cells from the mouse tumors. Data is shown below and in revised supplementary figure 5.

CD8T cells were gated in the PBMCs from the tumor cell-PBMC co-culture experiments as below.

Both PD1 and TIM3 expression are significantly reduced in CD8+ T cells after treatment with 6TdG. This data is included in revised manuscript figure 6.

9. What is the impact of co-culture between CD4 T cells and 6TdG-pretreated SCLC cells?

Response 9- As per the reviewer we isolated CD4 T cells from whole human PBMCs. We then cultured these cells with human SCLC cell lines (H69 or H510) pretreated with 6-TdG. There is increased release of IL-2 and IFN γ as compared CD4+ T cells co-cultured with untreated SCLC cells. Increase in IFN γ was only observed with one of the lines and this was a small difference. The data was added as supplementary Figure 4f.

10. Figure 6B: Are these absolute numbers for different subsets from single cell RNA-seq data or flow cytometry? It may not be ideal to use cell numbers to quantify differences in cell populations from scRNA-seq.

Response 10- We thank the reviewer for the recommendation. We removed this absolute data even though there was correlations with the absolute counts from flow, this was imperfect. We included tsNE plots for overview.

11. Which MDSC population in liver (Figure 6F) is impacted by 6TdG treatment- PMN-MDSC or M-MDSC?

Response 11- CD11b+Ly6C+Ly6G- monocytic MDSCs were impacted by 6TdG treatment. PMN-MDSCs (CD11b+Ly6G+) cells were not significantly affected. This statement was added to Result section, "6-TdG modulates the tumor microenvironment" subsection, 2nd paragraph. The figure demonstrating this was added as Supplementary figure 5a.

12. Figure 6F shows no difference in NK cell numbers after 6TdG treatment. Does 6TdG treatment only impact cytokine secretion from NK cells (as shown in co-culture experiment in supplementary Fig 2SB)? Did the authors also evaluate IFN γ from NK cells in in vivo model?

Response 12- In a new experiment with evaluated IFN gamma and CD107a expression among NK cells in 6TdG treated subcutaneous tumors. These data have been added as supplementary figure 5b in the revised manuscript.

13. In the in vivo model in Figure 6, what is the impact of treatment on T-cell activation (IFN γ , TNF α , granzyme B), proliferation (Ki67 and PD-1 expression)?

Response 13- IFN γ and granzyme b are increased indicating increased functional activation. This data is provided in response to comment # 8 of the reviewer.

14. What is the mechanism by which single agent 6TdG increase STING. Is there involvement of dsDNA, TREX etc.

Response 14- 6TdG treatment resulted in a significant increase in formation of micronuclei. A majority of these micronuclei were without a protective lamin layer as shown through uncoated telomeres, cytoplasmic DNA potentially activating cGAS. This data is shown in figure 1f-h and figure 5a in the revised manuscript. We are including data showing activation of cGAS/STING pathway and requirement of cGAS for type-I IFN production in tumor cells as below and in revised figure 5 and supplementary figure 3i.

15. It would be important to know if 6TdG is a better option to existing telomerase inhibitors, particularly in the context of combination with immunotherapy.

Response 15- Imetelstat did not induce cell death in any of the two human and two mouse SCLC lines *in vitro*. There was also not a significant therapeutic efficacy of imetelstat *in vivo*. This data is shown in supplementary figure 1b and 1c in the revised manuscript. Since we did not see single agent efficacy in mice, we therefore did not test imetelstat in combination with immunotherapy.

Imetelstat efficacy experiments

6TdG + PDL1 antibody *in vivo* experiment

We tested 6TdG in combination with PDL1 antibody in our 984 IV model. We see efficacy of both 6TdG and combination treatment however additional experiments are needed to confirm any potential synergies. Specifically, we need to test different dosing schedules to determine the best combinatory approach in future studies. Since 6TdG is effective alone we hypothesize there will be an additive affect with ICB therapy when given in the right setting.

Minor comments:

1. Details on co-culture experiments, how they were performed can be expanded in methods section?

Response 1- Detailed experiment protocols are included in the methods section which reads as: CD3, CD4, or whole human PBMCs were co-cultured with human cells lines at 10:1 ratio of immune cells to tumor cells. For CD3 and PBMC co-cultures T cells were activated with CD3 (1ug/mL) and CD28 (5ug/mL) antibody. Cells were cultured together for 48hr and harvested for flow cytometry or ELISA analysis.

2. For the radiation-6TdG combination therapy, please show a schema for tumor injection and treatment timeline in figure 7.

Response 2- Treatment schemas for tumor injection and treatment were added to revised figure 7.

3. Provide details on statistical analysis for every figure section individually in the figure legend.

Response 3- Figure legends were revised to include statistical analysis information.

4. Both B6129SF1/J and nu/nu mice are used with 984 model at different points in the study. Please label in the manuscript as well as figure legend what strain is used in a certain experiment.

Response 4- The figure legends were revised to include strain information. The mouse strain information was also labeled on the mouse schemas on the figures.

5. Minor grammatical errors:

a. Abstract "SCLC models that were dependent on immune cells"

The sentence was corrected: "We also observed dramatic immune-mediated synergy between 6TdG and irradiation in both syngeneic and humanized SCLC models."

b. Methods "B6129SF1/J mice or nu/nu mice"

These were corrected.

REVIEWERS' COMMENTS

Reviewer #1 (Remarks to the Author):

The authors have satisfactorily addressed my concerns

Reviewer #2 (Remarks to the Author):

I am not convinced by the additional data.

1. The TIF analysis shows a diffuse H2AX staining that indicates that at this level of damage specific sites can not be identified. The provided image shows that the author do not assay TIFs but overall damage.

Metaphase TIFs are explained by a cell cycle arrest. Please see literature by the Karlesder lab that demonstrated that cells arrested in M will inadvertently induce TIFs.

2. The TERT^{-/-} 984 cell experiment does not seem to be well controlled. Cells and tumors grow dramatically slower than the wt cells. This remains unexplained and challenges the overall interpretability of this experiment. Also, preexisting clonal variation in the 984 cells is not controlled for when generating 984 TERT KO cells.

Without these issues resolved my previous comments stand that the telomere effects are not demonstrated and I am not sure that this study does beyond revealing a general cytotoxic effect of the drug.

Reviewer #3 (Remarks to the Author):

Thanks for your careful attention to the critique. I am fully satisfied with the responses.

Response to comments from reviewer 2:

Comment 1:

I am not convinced by the additional data.

1. The TIF analysis shows a diffuse H2AX staining that indicates that at this level of damage specific sites can not be identified. The provided image shows that the author do not assay TIFs but overall damage.

Metaphase TIFs are explained by a cell cycle arrest. Please see literature by the Karlesder lab that demonstrated that cells arrested in M will inadvertently induce TIFs.

TIF experiment was not done on the metaphase cells. Metaphase spreads were for the chromosome analysis. We are showing additional images showing DNA damage only in 6TdG treated cells but not control cells so unlikely to be an artifact of the experiment.

We previously showed TIF formation is only observed with telomerase expressing lung fibroblast cells (BJ Tert) but not in BJ cells without telomerase.

Editorial Note:

Full citation: Ilgen Mender, Sergei Gryaznov, Z. Gunnur Dikmen, Woodring E. Wright, Jerry W. Shay; Induction of Telomere Dysfunction Mediated by the Telomerase Substrate Precursor 6-Thio-2'-Deoxyguanosine. *Cancer Discov* 1 January 2015; 5 (1): 82–95. <https://doi.org/10.1158/2159-8290.CD-14-0609>

Eglenen-Polat et al, response to reviewer #2

Induction of TIF foci was also observed with HT1080 colon cancer cells.

[Editorial Note: Figure redacted as no permission to be published could be obtained.]

Abdissalam et al, JBC, 2020

TIF formation was also previously validated in tumor tissues from the mouse treated with 6TdG.

Editorial Note:

Full citation:

Ilgem Mender, Anli Zhang, Zhenhua Ren, Chuanhui Han, Yafang Deng, Silvia Siteni, Huiyu Li, Jiankun Zhu, Aishwarya Vemula, Jerry W. Shay, Yang-Xin Fu,

Telomere Stress Potentiates STING-Dependent Anti-tumor Immunity,
Cancer Cell,

Volume 38, Issue 3,
2020,

Pages 400-411.e6,

ISSN 1535-6108,

<https://doi.org/10.1016/j.ccell.2020.05.020>.

<https://www.sciencedirect.com/science/article/pii/S1535610820302701>

An independent group led by Dr Maria Blasco also observed increased in TIF formation with 6TdG (6thio-dG, same molecule) treatment of mouse and human non-small lung cancer cells in a manuscript that came out within the last few months.

Mouse cells:

Human cells:

Editorial Note:

Full citation:

Piñero-Hermida, S., Bosso, G., Sánchez-Vázquez, R. et al. Telomerase deficiency and dysfunctional telomeres in the lung tumor microenvironment impair tumor progression in NSCLC mouse models and patient-derived xenografts. *Cell Death Differ* 30, 1585–1600 (2023). <https://doi.org/10.1038/s41418-023-01149-6>

Since TIF formation with 6TdG was extensively investigated, we moved the TIF figure to supplementary data and show individual colors for gH2AX, TRF2 and DAPI. We also revised the text as below:

Human SCLC cell line H841 was treated with 6TdG for 48 hours before staining. There was increase in both general double stranded breaks and telomere specific breaks indicated by g-H2AX accumulation colocalized with telomere probe Tel-C within the nucleus (Fig. S2a). The numbers of TIFs per cell were 31 times higher and g-H2AX- foci per cell were 19 times higher for cells treated with 6TdG as compared to control (Fig. S2b). Since telomeres are only 1/6000th of the genomic DNA, this data indicates preferential enrichment of damage at the telomeres.

Comment 2. The TERT^{-/-} 984 cell experiment does not seem to be well controlled. Cells and tumors grow dramatically slower than the wt cells. This remains unexplained and challenges the overall interpretability of this experiment. Also, preexisting clonal variation in the 984 cells is not controlled for when generating 984 TERT KO cells.

Without these issues resolved my previous comments stand that the telomere effects are not demonstrated and I am not sure that this study does beyond revealing a general cytotoxic effect of the drug.

Response: This experiment was actually done using a mixture of two independent homozygous Tert knockout clones of 984. We agree with the reviewer that these tumors grew slower. This experiment was an additional experiment we performed, not requested by the reviewer. We now

indicate in the results that these clones grew slower in vivo. Here we enclose additional data with another Tert knockout mouse model. Again, in this model (MC38), knockout cells grow slightly slower than Tert WT cells but are similarly resistant to 6TdG treatment. Collectively, we believe this data supports our conclusions that response to 6TdG in vivo is dependent on telomerase.

Unpublished